# R2R2: Robust Representation for Intensive Experience Reuse via Redundancy Reduction in Self-Predictive Learning

**Sanghyeob Song** [1][2]   **Donghyeok Lee** [1]   **Jinsik Kim** [3]   **Sungroh Yoon** [1][3]

## Abstract

For reinforcement learning in data-scarce domains like real-world robotics, intensive data reuse enhances efficiency but induces overfitting. While prior works focus on critic bias, representation-level instability in Self-Predictive Learning (SPL) under high Update-to-Data (UTD) regimes remains underexplored. To bridge this gap, we propose Robust Representation via Redundancy Reduction (R2R2), a regularization method within SPL. We theoretically identify that standard zero-centering conflicts with SPL's spectral properties and design a non-centered objective accordingly. We verify R2R2 on SPL-native algorithms like TD7. Furthermore, to demonstrate its orthogonality to prior advancements, we extend the state-of-the-art SimbaV2, which originally lacks SPL, by integrating a tailored SPL module, termed SimbaV2-SPL. Experiments across 11 continuous control tasks confirm that R2R2 effectively mitigates overfitting; specifically, at a UTD ratio of 20, it improves TD7 by ~22% and provides additional gains on top of SimbaV2-SPL, which itself establishes a new state-of-the-art. The code can be found at: https://github.com/songsang7/R2R2

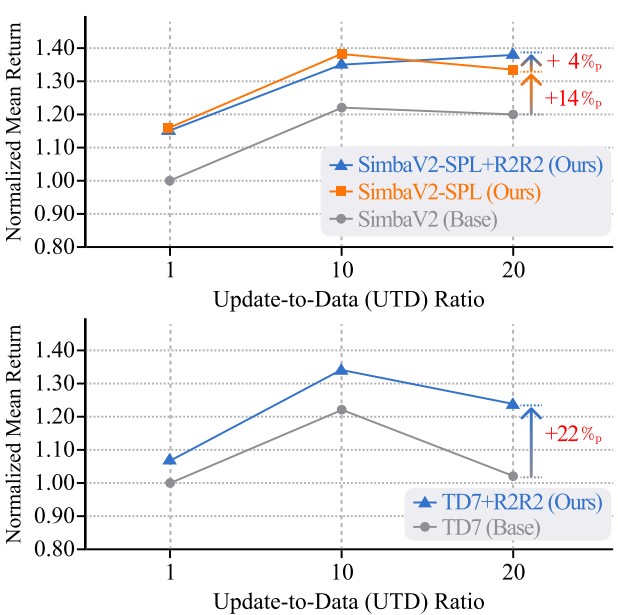

*Figure 1.* Performance comparison across UTD ratios 1, 10, and 20. Our method demonstrates robustness with minimal loss or gains in high UTD regimes. Notably, our proposed SimbaV2-SPL outperforms the current state-of-the-art, SimbaV2 (Lee et al., 2025b), and R2R2 achieves further performance gains on top of this enhanced baseline.

## 1. Introduction

Historically, reinforcement learning (RL) has evolved to improve sample efficiency. Off-policy methods (Mnih et al., 2015; Lillicrap et al., 2016; Fujimoto et al., 2018; Haarnoja et al., 2018a;b) have achieved this by reusing past transitions via an experience replay buffer, while model-based RL methods (Sutton, 1990; Janner et al., 2019; Ha & Schmidhuber, 2018) have done so by generating synthetic experiences from a learned model. Subsequently, Self-Predictive Learning (SPL) (Schwarzer et al., 2021; van den Oord et al., 2018) has emerged as a promising framework, employing an auxiliary task to predict the latent representation of the next state conditioned on the current state and action. SPL maximizes data efficiency by providing additional information about environmental dynamics for actor-critic training. Parallel to these advancements, another approach for improving sample efficiency is to intensively reuse collected experiences by increasing the Update-to-Data (UTD) ratio, defined as the number of gradient updates per environment interaction. To enable such intensive updates while mitigating overfitting, most prior works operate within the model-free paradigm, primarily focusing on addressing value estima-

[1]Interdisciplinary Program in Artificial Intelligence, Seoul National University [2]Samsung Electro-Mechanics, Republic of Korea [3]Department of Electrical and Computer Engineering, Seoul National University. Correspondence to: Sungroh Yoon <sryoon@snu.ac.kr>.

*Proceedings of the 43rd International Conference on Machine Learning*, Seoul, South Korea. PMLR 306, 2026. Copyright 2026 by the author(s).

tion bias. Prominent methods such as REDQ (Chen et al., 2021) and CrossQ (Bhatt et al., 2024) have successfully stabilized training by using ensembles or normalization.

However, these value-centric advancements primarily target the critic's output and do not explicitly address the representation-level overfitting. Consequently, the intersection of high UTD training and SPL remains underexplored. We argue that addressing representation-level instability is an orthogonal challenge to mitigating value bias; while high UTD regimes induce overfitting across all components, existing value-centric strategies are insufficient to prevent the representational degradation within the SPL encoder and latent dynamics model.

To address this, we propose Robust Representation via Redundancy Reduction (R2R2), a regularization approach incorporating redundancy reduction principles (Barlow et al., 1961; Zbontar et al., 2021; Bardes et al., 2022) to stabilize SPL performance across varying update frequencies, ranging from standard to high UTD settings (UTD $\approx 20$). Fundamentally, we diverge from standard redundancy reduction methods (Zbontar et al., 2021; Bardes et al., 2022) in computer vision by avoiding zero-centering. Building on the theoretical insight that SPL performs spectral decomposition of transition dynamics (Tang et al., 2023), we identify that centering mathematically eliminates the dominant eigenvector corresponding to the global dynamics information. Consequently, R2R2 employs a non-centered regularization scheme, preserving this information while ensuring robust feature learning even with intensive experience reuse. Crucially, since our approach targets the regularization of the encoder rather than the value function, our representation-level intervention is orthogonal to, and thus naturally compatible with, existing value-centric high UTD strategies.

To verify the efficacy and universality of R2R2, we first applied it to SPL-native algorithms such as TD7 (Fujimoto et al., 2023). On standard benchmarks, our method effectively mitigates performance degradation at high-UTD (UTD = 20), and improves the aggregate normalized score of TD7 by approximately 22%. This advantage is even more pronounced in DMC-Hard (Lee et al., 2025a), a challenging subset of the standard DMC suite (Tassa et al., 2018; 2020), where R2R2 significantly boosts the baseline performance (1.02 $\rightarrow$ 1.32). Furthermore, to demonstrate its orthogonality to architectural advancements, we targeted the state-of-the-art SimbaV2 (Lee et al., 2025b). Since our approach necessitates an SPL framework, we first integrated a tailored SPL module into this architecture, termed SimbaV2-SPL. This integration alone establishes a new state-of-the-art performance on continuous control benchmarks. Crucially, applying R2R2 to this enhanced baseline yields further gains, reaching a normalized score of 1.38, thereby confirming that our R2R2 provides complementary

benefits even to the strongest architectures.

In summary, the contributions of this paper are:

- **(Method)** We propose R2R2, a regularization approach grounded in our theoretical analysis. By leveraging redundancy reduction principles, our method explicitly mitigates representation-level instability caused by intensive experience reuse in high UTD regimes.

- **(Theory)** We provide a theoretical analysis identifying a fundamental conflict between the spectral decomposition properties of SPL and feature centralization (zero-centering). We demonstrate that zero-centering eliminates the dominant eigenmode representing global dynamics.

- **(Architecture)** We construct SimbaV2-SPL by integrating a tailored SPL framework into the state-of-the-art, SimbaV2. Distinct from our proposed regularization, this architectural extension bridges the gap between latent dynamics modeling and modern model-free backbones.

- **(Performance)** We demonstrate that R2R2 generally improves various backbone algorithms, particularly under high UTD regimes. Furthermore, our proposed architecture, SimbaV2-SPL, alone establishes a new state-of-the-art performance on continuous control benchmarks and on top of this enhanced baseline, SimbaV2-SPL +R2R2 achieves additional performance gains.

## 2. Related Works

### 2.1. Update-to-Data Ratio

Standard off-policy algorithms (Mnih et al., 2015; Lillicrap et al., 2016; Fujimoto et al., 2018; Haarnoja et al., 2018a;b) typically limit the Update-to-Data (UTD) ratio to 1 to avoid overfitting caused by intensive data reuse. However, recent trends favor high UTD ratios (e.g., UTD $\approx 20$). To mitigate this overfitting, Dyna-style approaches augment training with synthetic transitions (Sutton, 1990; Janner et al., 2019; Voelcker et al., 2025), while model-free methods utilize randomized ensembles (Chen et al., 2021) or successfully incorporate Batch Normalization (Bhatt et al., 2024; Ioffe & Szegedy, 2015) to mitigate bias. More recently, attention has shifted toward architectural innovations; SimBa (Lee et al., 2025a) and BRO (Nauman et al., 2024) demonstrate that regularization techniques like Layer Normalization (Ba et al., 2016) and Dropout (Srivastava et al., 2014) can rival model-based efficiency. Building on this, SimbaV2 (Lee et al., 2025b) further scaled network capacity and refined normalization, establishing a new state-of-the-art.

Despite these advancements, prior works primarily target instability through value function ensembles or architectural constraints. In contrast, the specific issue of representation-level degradation presents an orthogonal challenge that has received relatively limited attention.

## 2.2. Self-Supervised Learning

Self-Supervised Learning (SSL) has evolved from contrastive methods like SimCLR (Chen et al., 2020) to non-contrastive asymmetric approaches such as BYOL (Grill et al., 2020) and SimSiam (Chen & He, 2021). Distinct from these architectural solutions, Redundancy Reduction (Barlow et al., 1961)-based methods, including Barlow Twins (Zbontar et al., 2021) and VICReg (Bardes et al., 2022), prevent collapse by explicitly decorrelating feature dimensions. We leverage these principles to learn robust state representations in overfitting-prone RL scenarios.

## 2.3. Self-Predictive Learning

To enhance sample efficiency, prior works utilized auxiliary objectives ranging from reconstruction (Yarats et al., 2021; Jaderberg et al., 2017) to contrastive learning (van den Oord et al., 2018; Laskin et al., 2020). Moving beyond these, Self-Predictive Learning (SPL) focuses on capturing latent temporal dynamics rather than static features. Prominent implementations include SPR (Schwarzer et al., 2021), which adapts BYOL (Grill et al., 2020), and TD7 (Fujimoto et al., 2023), which employs a SimSiam (Chen & He, 2021)-style framework. Recently, theoretical studies (Tang et al., 2023; Ni et al., 2024) have further generalized the SPL framework, offering insights into its spectral properties.

# 3. Preliminaries

## 3.1. Reinforcement Learning Problem

We address the reinforcement learning problem within the framework of a Markov Decision Process (MDP), formalized as a tuple $(\mathcal{S}, \mathcal{A}, P, R, \gamma)$ (Sutton & Barto, 2018). The goal of an agent is to learn a policy $\pi$ that maximizes the expected discounted cumulative return. In the context of SPL, we focus on learning an informative latent representation $\phi : \mathcal{S} \to \mathbb{R}^k$ that encapsulates essential information about the transition dynamics $P$, which serves as a basis for sample-efficient learning.

## 3.2. Spectral Perspective on Self-Predictive Learning

SPL typically involves training an encoder $\phi$ alongside a latent transition predictor $\mathcal{T}$. The objective is to minimize the discrepancy between the predicted and actual future representations. Formally, for a transition tuple $(s_t, a_t, s_{t+1})$, the SPL loss ($\mathcal{L}_{\text{SPL}}$) is given by:

$$\mathcal{L}_{\text{SPL}} = \mathbb{E}_{(s_t, a_t, s_{t+1}) \sim \mathcal{D}} \left[ \|\mathcal{T}(\phi(s_t), a_t) - \text{sg}(\phi(s_{t+1}))\|_2^2 \right], \tag{1}$$

where $\text{sg}(\cdot)$ indicates the stop-gradient operation, a critical component for training stability.

**Connection to Spectral Decomposition.** Recent theoretical

advancements have established a link between the learning dynamics of SPL and the spectral decomposition of the state transition matrix. Consider the representation matrix $\Phi \in \mathbb{R}^{|\mathcal{S}| \times k}$, where each row corresponds to the embedding $\phi(s)$ of a state $s$. Specifically, (Tang et al., 2023) demonstrate that minimizing the SPL objective implicitly maximizes a trace functional involving the transition matrix $P^\pi$:

$$\max_{\Phi} \text{Tr} \left( (\Phi^\top P^\pi \Phi)^\top (\Phi^\top P^\pi \Phi) \right) \ \text{ s.t. } \ \Phi^\top \Phi = I_k. \tag{2}$$

The optimal representation $\Phi^*$ derived from this objective spans the subspace defined by the top-$k$ *right* eigenvectors of $P^\pi$. A fundamental property of a Markov chain with a row-stochastic transition matrix $P$ is that the eigenvalue 1 always exists. In particular, the constant vector $\mathbf{1}$ is a right eigenvector satisfying $P\mathbf{1} = \mathbf{1}$, reflecting conservation of probability mass. This constant eigenvector aligns with the global bias. This observation implies that preserving the constant mode is needed for capturing the global dynamics.

## 3.3. Redundancy Reduction in Self-Supervised Learning

To learn robust representations without collapse, redundancy reduction methods impose explicit constraints on the statistical properties of the embeddings. Among these approaches, VICReg (Bardes et al., 2022) is particularly effective as it decouples the learning objective into three independent regularization terms: Variance, Invariance, and Covariance.

For a formal definition, let $Z \in \mathbb{R}^{N \times d}$ denote a batch of feature vectors, where $N$ is the batch size and $d$ is the feature dimension. We denote $z_{b,\cdot}$ as the $b$-th sample vector and $z_{\cdot,j}$ as the vector of the $j$-th dimension across the batch. The components are defined as follows:

- **Variance ($\mathcal{L}_{\text{Var}}$):** Prevents representation collapse by maintaining the variance of each feature dimension above a threshold $v_{th}$, computed along the batch dimension:

$$\mathcal{L}_{\text{Var}}(Z) = \frac{1}{d} \sum_{j=1}^{d} \max(0, v_{th} - \sqrt{\text{Var}(z_{\cdot,j})}), \tag{3}$$

where $\text{Var}(z_{\cdot,j})$ is the variance of the $j$-th dimension.

- **Invariance ($\mathcal{L}_{\text{Inv}}$):** Minimizes the mean squared error between the representations of two views $(Z^A, Z^B)$ to learn invariant features against augmentations:

$$\mathcal{L}_{\text{Inv}}(Z^A, Z^B) = \frac{1}{N} \sum_{b=1}^{N} \|z_{b,\cdot}^A - z_{b,\cdot}^B\|_2^2. \tag{4}$$

- **Covariance ($\mathcal{L}_{\text{Cov}}$):** Decorrelates the feature dimensions by penalizing the off-diagonal coefficients of the covariance matrix. Crucially, the covariance matrix $\text{Cov}(Z)$ is typically centered:

$$\mathcal{L}_{\text{Cov}}(Z) = \frac{1}{d} \sum_{i \neq j} \left( \frac{1}{N-1} \sum_{b=1}^{N} (z_{b,i} - \mu_i)(z_{b,j} - \mu_j) \right)^2 \quad (5)$$

where $\mu_k = \mathbb{E}_b[z_{b,k}]$.

The total objective is a weighted sum: $\mathcal{L}_{\text{VIC}} = \lambda_{\text{Inv}}\mathcal{L}_{\text{Inv}} + \lambda_{\text{Var}}\mathcal{L}_{\text{Var}} + \lambda_{\text{Cov}}\mathcal{L}_{\text{Cov}}$, where each $\lambda$ denotes a balancing coefficient. Notably, the standard $\mathcal{L}_{\text{Cov}}$ relies on zero-centering, thereby treating the feature mean as a bias to be eliminated.

# 4. Method

In this section, we introduce our approach for robust representation learning in high UTD regimes. We start by analyzing the spectral conflict arises from applying conventional zero-centering to latent dynamics modeling (Sec. 4.1). This analysis reveals that the standard redundancy reduction, which enforces feature centering, is unsuitable for capturing environmental dynamics. Guided by this insight, we propose an objective refinements—such as omitting the projector and removing the centering constraint (Sec. 4.2). Furthermore, to extend our method to the state-of-the-art SimbaV2 (Lee et al., 2025b), which originally lacks an SPL component, we introduce an augmented architecture termed SimbaV2-SPL (Sec. 4.3).

## 4.1. Conflict between Zero-Centering and SPL

To formally demonstrate the conflict between SPL and zero-centering, we first define the batch-wise mean subtraction operation in matrix form and identify its property.

**Lemma 1** (Centering Matrix). *For a batch of size $N$, the zero-centering operation can be represented as a linear transformation by the centering matrix $H = I_N - \frac{1}{N}\mathbf{1}\mathbf{1}^\top$, where $I_N$ is the identity matrix and $\mathbf{1}$ is a column vector of ones. For any constant vector $\mathbf{c} = c\mathbf{1}$ (where $c \in \mathbb{R}$), the orthogonality condition holds:*

$$H\mathbf{c} = \mathbf{0}. \quad (6)$$

The detailed proof is provided in Appendix A.1.

This lemma implies that the centering operation removes any signal comprised of a constant component. Building on this, we show that centering explicitly eliminates the dominant eigenmode from the representation matrix $\Phi$, which is critical information captured by SPL.

**Proposition 2** (Elimination of Constant Eigenmode). *Let $\Phi^*$ be the optimal representation spanning the principal subspace of the row-stochastic transition matrix $P^\pi$. The zero-centering operation $H$ eliminates the projection of $\Phi^*$ onto the constant vector $u_1$ (corresponding to the dominant mode), i.e.,*

$$\|H\Phi^*_{\text{proj}, u_1}\|_2 = 0. \quad (7)$$

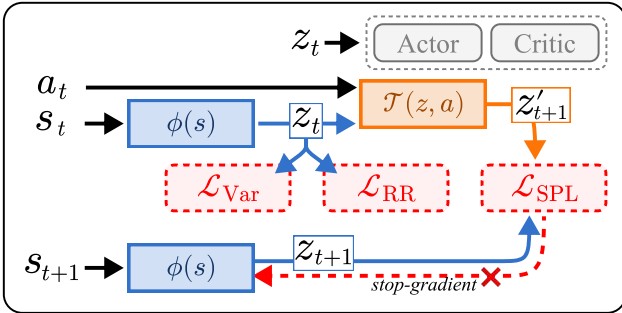

*Figure 2.* **Overview of R2R2.** The figure illustrates our proposed framework where direct regularization is applied to the latent representation $z_t$. This explicitly stabilizes feature learning.

The detailed proof is provided in Appendix A.2.

Proposition 2 suggests that even if the representation $\Phi$ successfully aligns with $u_1$ to capture the global information of transition dynamics, the application of zero-centering mathematically erases this information.

While the neural network's bias parameters could theoretically recover this eliminated component, relying on such implicit adaptation imposes an optimization disadvantage. To avoid this structural inefficiency, we adopt a non-centered regularization scheme that explicitly preserves the spectral information of SPL.

## 4.2. Dominant-Mode Preserving Regularization

Guided by the analysis in Sec. 4.1, we propose a modified redundancy reduction scheme that explicitly excludes mean subtraction to preserve the spectral information of SPL. Regarding the SPL loss, Eq. (1), we posit that an explicit redundancy reduction scheme based on VICReg (Bardes et al., 2022) is better suited to preserve the superior properties of the original SPL than Barlow Twins (Zbontar et al., 2021). Consequently, we introduce the following modifications to the original VICReg formulation:

**Non-centered Redundancy Reduction.** Instead of the standard covariance loss, we employ a non-centered inner product form to explicitly avoid eliminating the global information. We formally denote this objective as the Redundancy Reduction loss ($\mathcal{L}_{\text{RR}}$). It is computed using the non-centered correlation matrix in Eq. (8) and normalized by the total number of off-diagonal elements, $d(d-1)$, to ensure that the magnitude remains invariant to the feature dimension size.

$$\mathcal{L}_{\text{RR}} = \frac{1}{d(d-1)} \sum_{i \neq j} \left([C(Z)]_{ij}\right)^2, \quad (8)$$

where $[C(Z)]_{ij} = \frac{1}{N-1} \sum_{b=1}^{N} z_{b,i} z_{b,j}$ .

**Algorithm 1** Training Procedure of R2R2

Algorithm

1: **Input:** UTD ratio $G$, Batch size $N$, Regularization coefficients $\lambda_{\text{RR}}, \lambda_{\text{Var}}$
2: **Initialize:** Encoder $\phi$, Predictor $\mathcal{T}$, Actor $\pi$, Critic $Q$, Replay Buffer $\mathcal{D}$
3: **for** each environment step $t$ **do**
4:     Observe state $s_t$, select action $a_t \sim \pi(\phi(s_t))$
5:     Execute $a_t$, observe reward $r_t$, next state $s_{t+1}$
6:     Store transition $(s_t, a_t, r_t, s_{t+1})$ in $\mathcal{D}$
7:     ▷ High UTD Update Loop
8:     **for** $u = 1$ **to** $G$ **do**
9:         Sample batch $B \sim \mathcal{D}$
10:         ▷ 1. Self-Predictive Learning Block
11:         Encode states: $Z \leftarrow \phi(s),\ Z' \leftarrow \text{sg}(\phi(s'))$
12:         $\mathcal{L}_{\text{SPL}} \leftarrow$ SPL loss using $\mathcal{T}(Z, a)$ and $Z'$    Eq. (1)
13:         $\mathcal{L}_{\text{RR}} \leftarrow$ RR loss on $Z$                  Eq. (8)
14:         $\mathcal{L}_{\text{Var}} \leftarrow$ Variance loss on $Z$       Eq. (3)
15:         $\mathcal{L}_{\text{R2R2}} \leftarrow \mathcal{L}_{\text{SPL}} + \lambda_{\text{RR}}\mathcal{L}_{\text{RR}} + \lambda_{\text{Var}}\mathcal{L}_{\text{Var}}$   Eq. (9)
16:         Update Encoder $\phi$ and Predictor $\mathcal{T}$ using $\nabla \mathcal{L}_{\text{R2R2}}$
17:         ▷ 2. Reinforcement Learning Block (Base Algo.)
18:         Update Actor $\pi$ and Critic $Q$, using latent state $Z$ (following base algorithm)
19:     **end for**
20: **end for**

**Direct Regularization.** We apply regularization directly to the output of the encoder without an additional projector module. This design choice is grounded in our theoretical analysis, which provides a precise understanding of SPL dynamics, indicating that the auxiliary projector is redundant.

Based on these modifications, our final objective, $\mathcal{L}_{\text{R2R2}}$, explicitly enforces redundancy reduction, SPL-dynamics, and feature uniformity. (Eq. (9)):

$$\mathcal{L}_{\text{R2R2}}(Z) = \mathcal{L}_{\text{SPL}}(Z) + \lambda_{\text{RR}}\mathcal{L}_{\text{RR}}(Z) + \lambda_{\text{Var}}\mathcal{L}_{\text{Var}}(Z). \quad (9)$$

### 4.3. Architecture: SimbaV2-SPL

Our proposed regularization scheme is designed to be algorithm-agnostic, assuming only the existence of SPL, thereby offering universal applicability. To demonstrate this, we introduce SimbaV2-SPL, which integrates the tailored SPL framework into SimbaV2 (Lee et al., 2025b), the current state-of-the-art model in continuous control benchmarks. Since SimbaV2 is originally designed as a purely model-free architecture, it lacks an explicit mechanism to learn environmental dynamics. We address this by augmenting the architecture with an additional encoder and a transition predictor, trained specifically under the SPL framework. Specifically, rather than employing a generic encoder architecture typical of prior methods, such as TD7 (Fujimoto et al., 2023), we align our design with the distinct architectural philosophy of SimbaV2. Crucially, to ensure the preser-

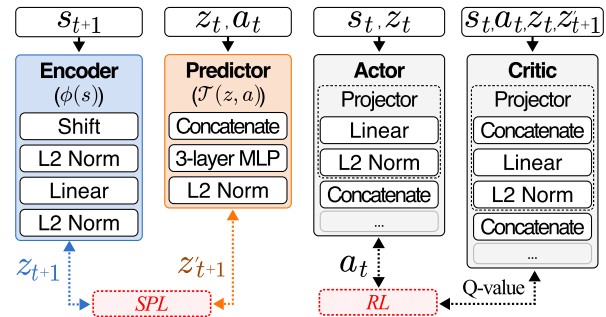

*Figure 3.* **SimbaV2-SPL.** We augment the backbone with a tailored SPL module (encoder $\phi$, predictor $\mathcal{T}$). The Actor and Critic networks are adapted to align with the SimbaV2 architecture, ensuring seamless integration of latent representations.

vation of raw information—particularly high-frequency details—that might be lost during encoding, we maintain the original state as a parallel input to the actor-critic networks, rather than entirely replacing it with the latent representation $z$. To integrate this input while adhering to SimbaV2's specific constraints, the state (or state-action pair) is first linearly projected and subsequently normalized using an $L_2$ norm. This processed input is then concatenated with the latent representation $z$. This integrated architecture, termed SimbaV2-SPL, serves as our enhanced baseline and is illustrated in Fig. 3.

## 5. Experiments

In this section, we validate the effectiveness and universality of our proposed method. We design our experiments to verify six key aspects: (1) robustness in high UTD regimes, (2) independence from algorithmic specifics, (3) distinctness from architectural normalization (Layer Normalization; Ba et al., 2016), (4) complementarity with state-of-the-art architectures, (5) the validity of our design choices via ablation studies, and (6) the spectral analysis of the learned representations through singular value spectrum.

**Environments.** We utilize a total of 11 environments. From OpenAI Gym MuJoCo (Brockman et al., 2016; Todorov et al., 2012), we select 4 environments: `Ant-v5`, `Walker2d-v5`, `Hopper-v5`, and `Humanoid-v5`. From the DeepMind Control Suite (DMC) (Tassa et al., 2018; 2020), we use the subset known as DMC-Hard (Lee et al., 2025a), which comprises 7 environments: `Humanoid-{Walk, Run, Stand}` and `Dog-{Trot, Walk, Stand, Run}`.

**Metric.** To capture sample efficiency, we measure the average return over the final 20% of training steps, following Machado et al. (2018). To allow for aggregation, we compute normalized scores relative to the corresponding baseline trained at UTD = 1 (see Appendix D for mathematical details). We report aggregated results here, while

individual task performance is provided in Appendix E.

**Baselines.** To rigorously evaluate the efficacy of our proposed regularization across varying levels of algorithmic complexity, we utilize the following baselines:

- **TD7 (Fujimoto et al., 2023):** A representative SPL-native algorithm, selected to demonstrate robustness within a standard latent-dynamics framework.

- **Minimalist $\phi$ (Ni et al., 2024):** A simplified SPL variant, chosen to verify that our performance gains are fundamental to the SPL objective rather than artifacts of complex auxiliary mechanisms.

- **TD7+Layer Normalization (TD7+LN) (Ba et al., 2016):** This variant incorporates Layer Normalization within the encoder to demonstrate that the performance improvements of R2R2 are distinct from and independent of standard architectural normalization.

- **SimbaV2 (Lee et al., 2025b):** The current state-of-the-art (SOTA) architecture in continuous control. Although it inherently lacks an SPL framework, we select it as the strongest available backbone. By integrating a tailored SPL module (termed SimbaV2-SPL), we aim to demonstrate that our method is orthogonal to architectural advancements and provides complementary gains.

We omit purely value-centric high-UTD algorithms, such as CrossQ (Bhatt et al., 2024), from direct comparison for two reasons. First, as discussed in Sec. 2, these methods address Q-function bias, which presents an orthogonal challenge to our representation-focused contribution. Second, since SimbaV2 has already demonstrated superior performance over these approaches, we select it as the representative state-of-the-art baseline.

**Training.** Training is conducted for a fixed budget of 500k decision steps. Notably, we fix the regularization coefficients $\lambda_{\text{Var}}$ and $\lambda_{\text{RR}}$ to $0.01$, and the variance threshold $v_{th}$ to 1 across all experiments without per-task tuning. All other hyperparameters follow the default settings of the base algorithms. For more details, see Appendix B.

### 5.1. Robustness in High UTD Regimes

Our primary hypothesis is that adding our regularization term to SPL-based algorithms enhances their robustness, mitigating potential degradation while unlocking further performance gains under high UTD settings. As shown in the TD7 (Fujimoto et al., 2023) section of Table 1 and Fig. 4-(TD7), R2R2 demonstrates remarkable robustness in the high UTD regime. While the TD7 baseline maintains a normalized score of 1.02 at UTD $= 20$, our method significantly boosts this to 1.24, achieving a 22% relative improvement. This gain is particularly pronounced in the complex DMC-Hard benchmark ($1.02 \rightarrow 1.32$), indicating

that the benefits of our regularization are magnified in challenging environments.

### 5.2. Independence from Algorithmic Specifics

We aim to demonstrate that the gains observed in TD7 (Fujimoto et al., 2023) are not artifacts of its complex auxiliary techniques but are fundamental to the SPL framework. As presented in the Minimalist $\phi$ (Ni et al., 2024) section of Table 1 and Fig. 4-(Minimalist $\phi$), results confirm that the benefits of R2R2 are fundamental. Even in the Minimalist $\phi$ setting, which lacks the sophisticated auxiliary mechanics of TD7, our method improves the Total aggregated mean from 5.28 to 6.20 at UTD $= 20$. Notably, in the Gym MuJoCo (Brockman et al., 2016; Todorov et al., 2012) benchmark where the baseline suffers severe collapse (dropping to 0.41), R2R2 effectively acts as a defense against such drastic drops, securing a score of 0.57. This suggests that our regularization enhances the robustness of the latent dynamics learning process, regardless of the algorithmic architecture. Additionally, at UTD $= 1$, combining the minimalist baseline with R2R2 still yields a clear improvement in performance ($1.00 \rightarrow 2.74$), indicating that our approach enriches the learned features even without intensive updates.

We note, however, that because the minimalist baseline attains very low absolute returns due to its minimal structure, the normalized improvement ratio can appear inflated when the reference score is small. To avoid overinterpreting this effect, we also provide the raw scores in the Appendix E.

### 5.3. Distinct Mechanism from Layer Normalization

A critical question is whether our approach operates via a mechanism distinct from standard architectural normalization. As presented in the TD7+LN (Fujimoto et al., 2023; Ba et al., 2016) section of Table 1 and Fig. 4-(TD7+LN), results reveal a compelling finding: architectural normalization alone is insufficient for high UTD robustness. The TD7+LN baseline, despite being a strong architectural variant, suffers from performance degradation at UTD $= 20$, with the Total aggregated mean dropping to 0.88 (falling below its UTD $= 1$ performance). In stark contrast, applying R2R2 on top of LN not only recovers from this degradation but further unlocks performance gains, reaching a score of 1.10. This significant recovery implies that R2R2 addresses a distinct form of representational collapse that Layer Normalization cannot resolve, thereby demonstrating the orthogonality of the two approaches.

### 5.4. Complementarity with Modern Architectures

We evaluated the compatibility of our method, R2R2, with SimbaV2 (Lee et al., 2025b), a state-of-the-art architecture featuring extensive normalization. As shown in Table 1 and

*Table 1.* **Aggregated normalized performance comparison.** We report the aggregated mean scores with 95% confidence intervals [Lower, Upper] across Gym MuJoCo (Brockman et al., 2016; Todorov et al., 2012) and DMC-Hard (Tassa et al., 2018; 2020) benchmarks. The Total column represents the aggregate score across all 11 environments. For SimbaV2 (Lee et al., 2025b), we explicitly show the performance gain from SPL and our proposed regularization method separately.

| | UTD = 1 (SANITY CHECK) | | | UTD = 20 (ROBUSTNESS) | | |
|---|---|---|---|---|---|---|
| ALGORITHM | GYM MUJOCO | DMC-HARD | TOTAL(4+7) | GYM MUJOCO | DMC-HARD | TOTAL(4+7) |
| **TD7** (FUJIMOTO ET AL., 2023) | | | | | | |
| BASE | 1.00 [0.96, 1.05] | 1.00 [0.83, 1.18] | 1.00 [0.88, 1.12] | 1.02 [0.93, 1.11] | 1.02 [0.84, 1.22] | 1.02 [0.89, 1.15] |
| + R2R2 (OURS) | **1.08** [1.03, 1.12] | **1.05** [0.86, 1.25] | **1.06** [0.93, 1.18] | **1.09** [1.03, 1.16] | **1.32** [1.14, 1.51] | **1.24** [1.12, 1.37] |
| **MINIMALIST** $\phi$ (NI ET AL., 2024) | | | | | | |
| BASE | 1.00 [0.90, 1.09] | 1.00 [0.79, 1.28] | 1.00 [0.87, 1.18] | 0.41 [0.24, 0.60] | 8.07 [2.56, 15.1] | 5.28 [1.75, 9.98] |
| + R2R2 (OURS) | 1.00 [0.87, 1.14] | **3.73** [0.84, 9.38] | **2.74** [0.89, 6.25] | **0.57** [0.38, 0.74] | **9.41** [3.30, 16.8] | **6.20** [2.03, 11.0] |
| **TD7+LN** (FUJIMOTO ET AL., 2023; BA ET AL., 2016) | | | | | | |
| BASE | 1.00 [0.93, 1.06] | 1.00 [0.89, 1.09] | 1.00 [0.93, 1.07] | 0.90 [0.78, 1.02] | 0.86 [0.78, 0.94] | 0.88 [0.80, 0.94] |
| + R2R2 (OURS) | 0.98 [0.93, 1.03] | **1.14** [1.03, 1.25] | **1.08** [1.01, 1.16] | **1.05** [0.97, 1.15] | **1.13** [1.02, 1.25] | **1.10** [1.03, 1.19] |
| **SIMBAV2** (LEE ET AL., 2025B) | | | | | | |
| BASE | 1.00 [0.95, 1.05] | 1.00 [0.92, 1.07] | 1.00 [0.95, 1.04] | 1.01 [0.95, 1.06] | 1.31 [1.17, 1.47] | 1.20 [1.10, 1.32] |
| + SPL (OURS) | **1.03** [0.97, 1.10] | 1.23 [1.15, 1.31] | **1.16** [1.09, 1.22] | **1.10** [1.04, 1.17] | 1.47 [1.24, 1.74] | 1.34 [1.18, 1.52] |
| + SPL + R2R2 (OURS) | 0.99 [0.93, 1.06] | **1.24** [1.15, 1.33] | 1.15 [1.08, 1.22] | 1.08 [1.00, 1.16] | **1.55** [1.26, 1.88] | **1.38** [1.19, 1.61] |

*Figure 4.* **Aggregated score curves of TD7 and R2R2.** Solid lines and shaded regions represent the mean and 95% confidence intervals, respectively. Our approach significantly outperforms the baseline at UTD=20 while maintaining comparable performance at UTD=1.

Fig. 4-(SimbaV2), integrating the tailored SPL framework alone (+SPL) already surpasses the baseline, establishing a new SOTA score of 1.34 at UTD = 20. Notably, the addition of our regularization (+SPL+R2R2) provides further gains, achieving a peak performance of 1.38.

While the margin of improvement (+0.04) may appear relatively modest compared to other baselines, we hypothesize that this is primarily due to a performance ceiling effect.

Given that the SimbaV2(+SPL) backbone is already exceptionally strong across several environments, the room for further absolute gains is fundamentally limited. Nevertheless, R2R2 provides a distinct, complementary benefit by preventing representational collapse. As substantiated by our subsequent Effective Rank analysis (Section 5.7), preserving this structural integrity contributes to further performance gains even in highly saturated regimes.

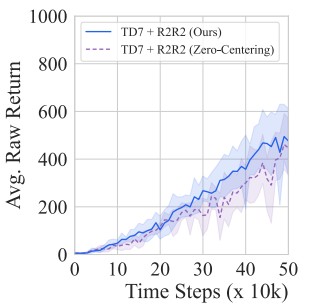
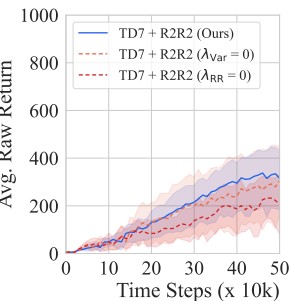

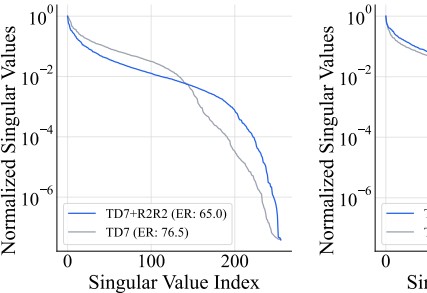
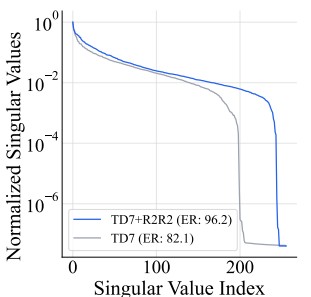

*Figure 5.* **Ablation Analysis. (Left)** Performance comparison regarding the zero-centering constraint. The results demonstrate that enforcing zero-centering leads to performance degradation, confirming the importance of preserving global information. **(Right)** Component-wise analysis verifying the contribution of individual loss terms to the final performance.

*Figure 6.* **The singular value spectrum. (Left)** At UTD $=$ 1, R2R2 shows a steeper initial decay. **(Right)** At UTD $=$ 20, baseline suffers from spectral cutoff in the tail indices.

## 5.5. Ablation Study

We conduct two ablation studies on the `Dog-Trot` environment from DMC (Tassa et al., 2018; 2020) to comprehensively validate our design choices. We chose this high-dimensional environment as it represents one of the most challenging tasks.

**Impact of Zero-Centering Constraint.** As shown in Fig. 5-(Left), comparing our framework with a zero-centered variant reveals that enforcing zero-centering degrades performance. This constraint strips away principal components, forcing the neural network to inefficiently reconstruct lost information at each step. In contrast, our non-centered design, $\mathcal{L}_{\text{RR}}$, yields better performance by preserving these essential features.

**Contribution of Regularization Terms.** We assess the individual roles of $\mathcal{L}_{\text{Var}}$ and $\mathcal{L}_{\text{RR}}$ by ablating each term at UTD $=$ 20. Fig. 5-(Right) confirms that both terms are complementary. While both ablations lead to performance degradation, the absence of $\mathcal{L}_{\text{RR}}$ causes a more severe drop. This validates that simultaneously enforcing variance preservation and decorrelating feature dimensions are critical for achieving robust learning in high UTD settings.

## 5.6. Spectral Analysis

To investigate R2R2, we analyze the singular value spectrum using TD7 (Fujimoto et al., 2023) as the base algorithm, focusing on `Humanoid-Stand` as it exhibits one of the most pronounced divergence between methods, thereby facilitating clear observation. Ideal representations exhibit a power-law decay: a steep initial drop indicates efficient compression of global dynamics, while a heavy tail ensures the preservation of fine-grained local variations without subspace collapse. Fig. 6 confirms that our regularization optimizes the latent structure towards this ideal profile.

**Spectral Concentration at UTD $=$ 1.** In the standard

regime, R2R2 shows a steeper initial decay compared to the baseline, resulting in a lower Effective Rank (ER $\approx$ 65.0 vs. 76.5). This indicates spectral concentration, where the model filters out redundant signals to compress task-relevant information into a compact set of principal components.

**Structural Integrity at UTD $=$ 20.** Conversely, in the high UTD regime, the baseline suffers from a sharp spectral cutoff, with singular values in the tail indices dropping rapidly to near-zero. This suggests a partial collapse where the model loses the capacity to capture fine-grained features. In contrast, R2R2 maintains a heavy-tailed distribution, confirming that our method effectively prevents subspace collapse, ensuring the full utilization of the latent space to represent diverse dynamics.

## 5.7. Effective Rank Monitoring

To further clarify the representation instability discussed in previous sections, we monitor the evolution of the Effective Rank (ER) throughout the training process. To highlight the divergence most clearly, we conduct this analysis under the extreme setting of UTD $=$ 20 on the `Humanoid-Run` environment, where our method demonstrated substantial performance gains.

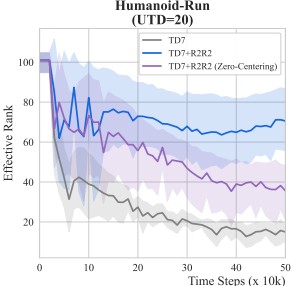
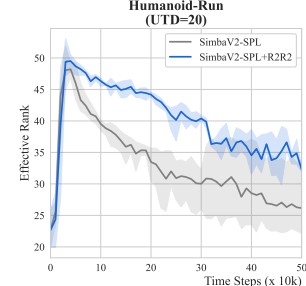

*Figure 7.* **Effective Rank (ER) over training steps.** Evaluated on `Humanoid-Run` at UTD $=$ 20. **(Left)** Comparison among the TD7 baseline, R2R2, and R2R2 with the zero-centering constraint. **(Right)** Comparison on the SimbaV2+SPL backbone with and without R2R2, demonstrating complementarity.

Fig. 7-(Left) tracks the ER for the TD7 baseline and our variants. While the unregularized baseline suffers a progressive loss of dimensionality, R2R2 successfully maintains a stable and high ER. Furthermore, when the zero-centering constraint is applied alongside our regularization, the ER drops progressively, closely mirroring the collapse trajectory of the baseline. This confirms that preserving non-centered features is crucial for preventing subspace collapse.

To substantiate the complementarity of our method with modern architectures, we also analyze the ER on the state-of-the-art SimbaV2+SPL backbone. As shown in Fig. 7-(Right), even though SimbaV2 is equipped with extensive architectural normalizations, the integration of R2R2 still yields a noticeably higher and more stable ER throughout the learning process. This empirical evidence clearly demonstrates that R2R2 and architectural improvements contribute to representation learning in fundamentally different ways; while SimbaV2 provides a strong structural capacity, R2R2 explicitly defends against representational collapse, acting as a highly complementary component for robust learning.

### 5.8. Training Time

We measure the wall-clock training time (sec/step) on `Humanoid-v5` at UTD = 20. This setting represents a worst-case scenario for computational cost: the high observation dimension maximizes the overhead of our method, while the frequent updates ensure this cost dominates the environment simulation time. We conduct all measurements on an Intel i7-9800X and an NVIDIA RTX2080Ti. Note that we implemented all SimbaV2 (Lee et al., 2025b) variants on top of the official JAX (Bradbury et al., 2018) implementation (Lee et al., 2025b), whereas TD7 (Fujimoto et al., 2023) utilizes PyTorch (Paszke et al., 2019).

*Table 2.* **Measurements of training time.** Average wall-clock training time (sec/step) measured with UTD = 20.

| METHOD | TD7 | SIMBAV2 |
|---|---|---|
| BASE | 0.417 | 0.210 |
| OURS | 0.465 | 0.235 (+SPL) |
| | | 0.241 (+SPL +R2R2) |

## 6. Conclusion

In this work, we identified a fundamental conflict where standard zero-centering undermines SPL by eliminating the principal spectral mode. To resolve this, we proposed R2R2, a non-centered redundancy reduction term designed to preserve this information. Our evaluation validates R2R2 across four dimensions: 1) **Robustness**: It mitigates high-UTD degradation, significantly boosting the TD7 (Fujimoto et al., 2023) baseline. 2) **Independence**: Confirmed via Minimalist $\phi$ (Ni et al., 2024), verifying efficacy independent of auxiliary algorithmic components. 3) **Distinctness**:

It addresses instability issues that Layer Normalization (Ba et al., 2016) cannot. 4) **Complementarity**: We constructed a new baseline, termed SimbaV2-SPL, by enhancing the current SOTA, SimbaV2 (Lee et al., 2025b). Notably, R2R2 achieves additional gains even on top of this robust architecture (SimbaV2-SPL +R2R2). Furthermore, ablation studies and spectral analysis provide additional verification of our design choices and theoretical soundness.

## 7. Limitations and Future Work

While our method effectively mitigates degradation at high UTD, it does not fully eliminate instability in extreme regimes. Another limitation is that, although we performed partial validation on pixel-based visual RL during the review process, we have not yet carried out a full-scale evaluation in that domain. Since the current study mainly focuses on low-dimensional states, extending and thoroughly validating the method in visual RL, where feature statistics are more complex, remains an important direction for future work.

## Acknowledgements

This work was supported by Samsung Electro-Mechanics.

This work was supported by Institute of Information & communications Technology Planning & Evaluation (IITP) grant funded by the Korea government(MSIT) [NO.RS-2021-II211343, Artificial Intelligence Graduate School Program (Seoul National University)]

This work was supported by the National Research Foundation of Korea (NRF) grant funded by the Korea government (MSIT) (No. 2022R1A3B1077720, 2022R1A5A7083908)

This work was supported by the BK21 FOUR program of the Education and Research Program for Future ICT Pioneers, Seoul National University in 2026.

## Impact Statement

This paper presents work whose goal is to advance the field of Machine Learning. There are many potential societal consequences of our work, none which we feel must be specifically highlighted here.

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

# A. Proofs

In this section, we provide detailed proofs for the conflict between zero-centering and the spectral properties of Self-Predictive Learning (SPL), as discussed in Section 4.1.

## A.1. Proof of Lemma 1

*Proof.* Recall that the centering matrix is defined as $H = I_N - \frac{1}{N}\mathbf{1}\mathbf{1}^\top$, where $I_N$ is the identity matrix of size $N$, and $\mathbf{1} \in \mathbb{R}^N$ is a column vector of ones. Let $\mathbf{c} = c\mathbf{1}$ be an arbitrary constant vector with a scalar $c \in \mathbb{R}$.

We compute the matrix-vector product $H\mathbf{c}$ as follows:

$$
\begin{aligned}
H\mathbf{c} &= \left( I_N - \frac{1}{N}\mathbf{1}\mathbf{1}^\top \right)(c\mathbf{1}) \\
&= c\mathbf{1} - \frac{c}{N}\mathbf{1}(\mathbf{1}^\top\mathbf{1}).
\end{aligned}
\tag{10}
$$

Since $\mathbf{1}$ is a vector of ones, the dot product $\mathbf{1}^\top\mathbf{1}$ sums to the dimension $N$:

$$
\mathbf{1}^\top\mathbf{1} = \sum_{i=1}^{N} 1 \cdot 1 = N.
\tag{11}
$$

Substituting this back into the equation:

$$
\begin{aligned}
H\mathbf{c} &= c\mathbf{1} - \frac{c}{N}\mathbf{1}(N) \\
&= c\mathbf{1} - c\mathbf{1} \\
&= \mathbf{0}.
\end{aligned}
\tag{12}
$$

Thus, applying the centering matrix $H$ to any constant vector results in the zero vector. This formally demonstrates that the centering operation removes the constant component (DC component) from any signal. $\square$

## A.2. Proof of Proposition 2

*Proof of Proposition 2.* We build upon the theoretical framework of Tang et al. (Tang et al., 2023). Their analysis implies that maximizing the SPL objective implicitly maximizes

$$
\mathrm{Tr}\left( (\Phi^\top P^\pi \Phi)^\top (\Phi^\top P^\pi \Phi) \right) \quad \text{subject to} \quad \Phi^\top \Phi = I_K,
\tag{13}
$$

and therefore the optimal representation $\Phi^*$ spans a principal subspace corresponding with the top-$K$ eigenmodes of $P^\pi$.

To make the argument precise, we will use the following lemma (Lemma 3), which characterizes the leading eigenvalue/eigenvector of a row-stochastic transition matrix. In particular, it allows us to choose the principal eigenvalue as $\lambda_1 = 1$ with corresponding eigenvector proportional to the constant vector $\mathbf{1}$.

**Lemma 3** (Spectral radius of a row-stochastic matrix). *Let $A \in \mathbb{R}^{n \times n}$ be* row-stochastic, *i.e., $A_{ij} \geq 0$ for all $i, j$ and $\sum_{j=1}^{n} A_{ij} = 1$ for all $i$. Then:*

*1. $1$ is an eigenvalue of $A$ with right eigenvector $\mathbf{1}$, i.e. $A\mathbf{1} = \mathbf{1}$.*

*2. Every eigenvalue $\lambda$ of $A$ satisfies $|\lambda| \leq 1$. In particular, no eigenvalue satisfies $\lambda > 1$.*

*Proof.* (1) Since each row of $A$ sums to 1, for every $i$,

$$
(A\mathbf{1})_i = \sum_{j=1}^{n} A_{ij} \cdot 1 = 1,
\tag{14}
$$

hence $A\mathbf{1} = \mathbf{1}$.

(2) Let $x \neq 0$ and define $M := \|x\|_\infty = \max_j |x_j|$. For each coordinate,

$$|(Ax)_i| = \left| \sum_{j=1}^n A_{ij} x_j \right| \leq \sum_{j=1}^n A_{ij} |x_j| \leq \sum_{j=1}^n A_{ij} M = M, \tag{15}$$

where we used $A_{ij} \geq 0$ and $\sum_j A_{ij} = 1$. Thus $\|Ax\|_\infty \leq \|x\|_\infty$. If $Ax = \lambda x$, then

$$|\lambda| \, \|x\|_\infty = \|\lambda x\|_\infty = \|Ax\|_\infty \leq \|x\|_\infty, \tag{16}$$

so $|\lambda| \leq 1$. In particular, no eigenvalue satisfies $\lambda > 1$. $\qquad\square$

Now we complete the proof of Proposition 2. Let $\{(\lambda_i, u_i)\}$ be eigenpairs of $P^\pi$, ordered by $|\lambda_1| \geq |\lambda_2| \geq \cdots$. Since $P^\pi$ is row-stochastic, by Lemma 3 we have $\lambda_1 = 1$ and we may choose the corresponding right eigenvector as the constant vector $u_1 = \alpha \mathbf{1}$ for some $\alpha \neq 0$.

Define the orthogonal projector onto $\mathrm{span}(u_1)$ by

$$\Pi_{u_1} := \frac{u_1 u_1^\top}{u_1^\top u_1}. \tag{17}$$

Using this, define the component of the learned representation $\Phi^*$ along $u_1$ as

$$\Phi^*_{\mathrm{proj}, u_1} := \Pi_{u_1} \Phi^*. \tag{18}$$

Let $H$ denote the zero-centering matrix (Lemma 1), so that $H\mathbf{1} = \mathbf{0}$. Since $u_1 = \alpha \mathbf{1}$, we have $Hu_1 = \alpha H\mathbf{1} = \mathbf{0}$, and hence

$$H\Pi_{u_1} = H \frac{u_1 u_1^\top}{u_1^\top u_1} = \frac{(Hu_1) u_1^\top}{u_1^\top u_1} = 0. \tag{19}$$

Therefore,

$$H\Phi^*_{\mathrm{proj}, u_1} = H(\Pi_{u_1} \Phi^*) = (H\Pi_{u_1})\Phi^* = 0,$$

and consequently,

$$\|H\Phi^*_{\mathrm{proj}, u_1}\|_2 = 0. \tag{20}$$

This proves that the zero-centering operation mathematically annihilates the component of the representation corresponding to the global dynamics (the constant eigenmode), thereby leading to a loss of spectral information. $\qquad\square$

# B. Hyperparameters

For baseline algorithms, including TD7 (Fujimoto et al., 2023), Minimalist $\phi$ (Ni et al., 2024), and SimbaV2 (Lee et al., 2025b), we adopt the default configurations provided in their respective original implementations, unless explicitly overridden by the common settings listed below. This ensures a fair comparison across all methods. All parameters are fixed across all 11 environments (4 Gym MuJoCo (Brockman et al., 2016; Todorov et al., 2012) and 7 DMC-Hard (Tassa et al., 2018; 2020; Lee et al., 2025a) tasks).

## B.1. Common Hyperparameters

Table 3 lists the shared hyperparameters applied uniformly across all agents to ensure consistent evaluation conditions. Note that for Gym MuJoCo (Brockman et al., 2016; Todorov et al., 2012), decision steps are equivalent to environment steps. In contrast, for DMC-Hard (Tassa et al., 2018; 2020) tasks, which utilize an action repeat of 2, our training budget of 500,000 decision steps corresponds to 1,000,000 environment steps.

## B.2. SimbaV2-SPL Specific Hyperparameters

For the backbone architecture and general optimization details, we strictly adhere to the original configurations of SimbaV2 (Lee et al., 2025b). Table 4 lists only the additional parameters introduced for the tailored SPL framework.

Table 3. Common Hyperparameters for Training.

| PARAMETER | VALUE |
|---|---|
| **GENERAL SETTINGS** | |
| TOTAL DECISION STEPS | 500,000 |
| ACTION REPEAT (GYM) | 1 |
| ACTION REPEAT (DMC) | 2 |
| NUMBER OF SEEDS | 5 |
| **REGULARIZATION** | |
| $\lambda_{\text{RR}}$ | 0.01 |
| $\lambda_{\text{VAR}}$ | 0.01 |
| $v_{th}$ | 1.0 |

Table 4. SimbaV2-SPL specific Hyperparameters.

| PARAMETER | VALUE |
|---|---|
| **NETWORK ARCHITECTURE** | |
| ENCODER($\phi$) WIDTH | 128 |
| ENCODER($\phi$) DEPTH | 1 |
| PREDICTOR($\mathcal{T}$) WIDTH | 256 |
| PREDICTOR($\mathcal{T}$) DEPTH | 3 |
| **OPTIMIZATION & INIT.** | |
| SPL LEARNING RATE (INIT) | $3 \times 10^{-4}$ |
| SPL LEARNING RATE (END) | $5 \times 10^{-5}$ |
| ENCODER C-SHIFT | 3.0 |

# C. Environments

Table 5. **Environment specifications.** We list the observation and action dimensions for the Gym MuJoCo (Brockman et al., 2016; Todorov et al., 2012) and DMC-Hard (Tassa et al., 2018; 2020) benchmarks.

| ENVIRONMENT | OBSERVATION DIMENSION | ACTION DIMENSION |
|---|---|---|
| **GYM MUJOCO** | | |
| ANT-V5 | 105 | 8 |
| WALKER2D-V5 | 17 | 6 |
| HOPPER-V5 | 11 | 3 |
| HUMANOID-V5 | 348 | 17 |
| **DMC-HARD** | | |
| HUMANOID-WALK | 67 | 21 |
| HUMANOID-RUN | 67 | 21 |
| HUMANOID-STAND | 67 | 21 |
| DOG-TROT | 223 | 38 |
| DOG-WALK | 223 | 38 |
| DOG-STAND | 223 | 38 |
| DOG-RUN | 223 | 38 |

# D. Evaluation Metric Details

In this section, we provide the mathematical definition of the metrics used in our evaluation. Let $R(i, k)$ denote the evaluation return at the $k$-th checkpoint for the $i$-th random seed. Given a total of $K = 50$ uniformly distributed checkpoints, we focus on the final 20% of training, which corresponds to the last $M = 10$ checkpoints.

**Average Return.** The temporal mean return $S(i)$ for a specific seed $i$ is defined as:

$$S(i) = \frac{1}{M} \sum_{k=K-M+1}^{K} R(i, k) \tag{21}$$

**Normalized Score.** To aggregate results across environments with varying reward scales and to robustly **handle potentially negative returns**, we compute the normalized score $\hat{S}(i)$. We define the normalization relative to the aggregate performance of the baseline algorithm. The normalized score is calculated as:

$$\hat{S}(i) = 1 + \frac{S(i) - \mathbb{E}_j \left[ S_{\text{base}}(j) \right]}{|\mathbb{E}_j \left[ S_{\text{base}}(j) \right]|} \tag{22}$$

where $\mathbb{E}_j[\cdot]$ denotes the empirical expectation (average) over the baseline seeds $j \in \{1, \ldots, N\}$. Here, the subscript "base" refers to the baseline algorithm trained at UTD $= 1$. Consequently, $S_{\text{base}}(j)$ represents the average return of the baseline's $j$-th seed, computed using the same protocol as in Eq. (21):

$$S_{\text{base}}(j) = \frac{1}{M} \sum_{k=K-M+1}^{K} R_{\text{base}}(j, k) \tag{23}$$

where $R_{\text{base}}(j, k)$ denotes the evaluation return of the baseline algorithm (trained at UTD $= 1$) at the $k$-th checkpoint for the $j$-th seed.

# E. Detailed Experimental Results

This section presents the complete set of experimental results. Before detailing the full tables and figures, we address a key statistical observation regarding the performance metrics.

**Metric.** To ensure a rigorous evaluation, we report the Interquartile Mean (IQM) (Agarwal et al., 2021) alongside the standard mean. As shown in the subsequent tables, while R2R2 consistently outperforms the baselines in IQM, the performance gap is notably wider in the standard mean. This discrepancy suggests that our performance gains stem primarily from improved robustness. The mean metric is sensitive to outliers, including catastrophic failures where agents suffer from representation collapse in high UTD regimes. Since IQM discards the bottom 25% of the data distribution, it effectively masks these instability issues inherent in the baselines. **In contrast, by incorporating our proposed regularization term into the loss function, R2R2 effectively mitigates these failures and significantly improves the performance of "worst-case" seeds.** Consequently, the larger gain in Mean (which accounts for preventing failures) compared to IQM (which ignores them) confirms that our regularization acts as a crucial "safety net," enhancing the overall reliability of the algorithm.

**Common Observations across Baselines.** As shown in all tables below, R2R2 yields scores comparable to or higher than the baseline across most tasks at UTD $= 1$. However, we observe a performance drop in some high-dimensional environments like `Humanoid-{Walk, Run}`. This reflects a characteristic trade-off: the regularization terms ($\mathcal{L}_{\text{Var}}$ and $\mathcal{L}_{\text{RR}}$) actively enforce feature diversity, preventing premature convergence. While this may slightly delay policy specialization in standard regimes (UTD $= 1$), it acts as a critical stabilizer in data-intensive settings. Indeed, at UTD $= 20$, R2R2 surpasses the baseline even in these tasks, confirming that **the benefits of regularization outweigh the initial cost.**

## E.1. TD7 baseline

*Table 6.* Raw score performance comparison on TD7 (Base vs. Ours) at UTD = 1, UTD = 10, and UTD = 20. We report mean $\pm$ std.

| | ENVIRONMENT | UTD = 1 | | UTD = 10 | | UTD = 20 | |
| | | TD7 (BASE) | TD7 (OURS) | TD7 (BASE) | TD7 (OURS) | TD7 (BASE) | TD7 (OURS) |
|---|---|---|---|---|---|---|---|
| GYM MUJOCO | ANT-V5 | $6897.9 \pm 441.0$ | $\mathbf{7890.2} \pm \mathbf{351.5}$ | $7170.0 \pm 751.7$ | $\mathbf{7981.4} \pm \mathbf{607.7}$ | $6863.3 \pm 1928.7$ | $\mathbf{7623.0} \pm \mathbf{516.7}$ |
| | WALKER2D-V5 | $5335.4 \pm 691.3$ | $\mathbf{5477.8} \pm \mathbf{543.5}$ | $\mathbf{5792.3} \pm \mathbf{815.5}$ | $5532.2 \pm 729.7$ | $6034.4 \pm 413.5$ | $\mathbf{6101.7} \pm \mathbf{59.0}$ |
| | HOPPER-V5 | $2776.4 \pm 427.5$ | $\mathbf{2911.2} \pm \mathbf{468.7}$ | $3088.2 \pm 575.2$ | $\mathbf{3238.5} \pm \mathbf{429.9}$ | $3281.9 \pm 340.9$ | $\mathbf{3529.0} \pm \mathbf{80.3}$ |
| | HUMANOID-V5 | $5262.5 \pm 459.8$ | $\mathbf{5787.6} \pm \mathbf{458.3}$ | $4753.1 \pm 508.6$ | $\mathbf{4917.7} \pm \mathbf{391.4}$ | $4092.3 \pm 443.2$ | $\mathbf{4463.7} \pm \mathbf{188.8}$ |
| DMC-HARD | HUMANOID-WALK | $\mathbf{279.4} \pm \mathbf{253.1}$ | $118.0 \pm 162.2$ | $527.6 \pm 31.3$ | $\mathbf{587.5} \pm \mathbf{48.7}$ | $462.4 \pm 80.3$ | $\mathbf{493.6} \pm \mathbf{96.9}$ |
| | HUMANOID-RUN | $\mathbf{74.6} \pm \mathbf{65.7}$ | $49.1 \pm 65.2$ | $\mathbf{146.8} \pm \mathbf{26.1}$ | $132.9 \pm 75.9$ | $143.5 \pm 10.8$ | $\mathbf{162.5} \pm \mathbf{33.4}$ |
| | HUMANOID-STAND | $411.3 \pm 382.6$ | $\mathbf{763.2} \pm \mathbf{160.2}$ | $669.6 \pm 108.8$ | $\mathbf{828.2} \pm \mathbf{30.7}$ | $375.6 \pm 247.2$ | $\mathbf{640.3} \pm \mathbf{170.7}$ |
| | DOG-TROT | $375.5 \pm 74.4$ | $\mathbf{453.1} \pm \mathbf{139.3}$ | $343.3 \pm 129.5$ | $\mathbf{425.4} \pm \mathbf{42.2}$ | $206.7 \pm 40.1$ | $\mathbf{322.6} \pm \mathbf{160.3}$ |
| | DOG-WALK | $699.2 \pm 103.4$ | $\mathbf{808.7} \pm \mathbf{45.6}$ | $663.4 \pm 134.5$ | $\mathbf{757.2} \pm \mathbf{124.9}$ | $282.5 \pm 128.2$ | $\mathbf{610.2} \pm \mathbf{82.4}$ |
| | DOG-STAND | $\mathbf{902.8} \pm \mathbf{23.4}$ | $872.8 \pm 88.0$ | $853.7 \pm 63.8$ | $\mathbf{932.1} \pm \mathbf{23.2}$ | $816.1 \pm 44.7$ | $\mathbf{872.3} \pm \mathbf{50.8}$ |
| | DOG-RUN | $205.6 \pm 32.6$ | $\mathbf{228.5} \pm \mathbf{52.7}$ | $201.7 \pm 23.6$ | $\mathbf{265.4} \pm \mathbf{35.1}$ | $159.6 \pm 37.7$ | $\mathbf{214.9} \pm \mathbf{26.4}$ |

*Table 7.* Performance comparison on TD7 (Base vs. Ours) at UTD = 1, UTD = 10, and UTD = 20. We report normalized mean $\pm$ std. The final rows report the aggregate Mean and IQM with 95% CIs.

| | ENVIRONMENT | UTD = 1 | | UTD = 10 | | UTD = 20 | |
| | | TD7 (BASE) | TD7 (OURS) | TD7 (BASE) | TD7 (OURS) | TD7 (BASE) | TD7 (OURS) |
|---|---|---|---|---|---|---|---|
| GYM MUJOCO | ANT-V5 | $1.00 \pm 0.06$ | $\mathbf{1.14} \pm \mathbf{0.05}$ | $1.04 \pm 0.11$ | $\mathbf{1.16} \pm \mathbf{0.09}$ | $1.00 \pm 0.28$ | $\mathbf{1.11} \pm \mathbf{0.08}$ |
| | WALKER2D-V5 | $1.00 \pm 0.13$ | $\mathbf{1.03} \pm \mathbf{0.10}$ | $\mathbf{1.09} \pm \mathbf{0.15}$ | $1.04 \pm 0.14$ | $1.13 \pm 0.08$ | $\mathbf{1.14} \pm \mathbf{0.01}$ |
| | HOPPER-V5 | $1.00 \pm 0.15$ | $\mathbf{1.05} \pm \mathbf{0.17}$ | $1.11 \pm 0.21$ | $\mathbf{1.17} \pm \mathbf{0.16}$ | $1.18 \pm 0.12$ | $\mathbf{1.27} \pm \mathbf{0.03}$ |
| | HUMANOID-V5 | $1.00 \pm 0.09$ | $\mathbf{1.10} \pm \mathbf{0.09}$ | $0.90 \pm 0.10$ | $\mathbf{0.93} \pm \mathbf{0.07}$ | $0.78 \pm 0.08$ | $\mathbf{0.85} \pm \mathbf{0.04}$ |
| DMC-HARD | HUMANOID-WALK | $\mathbf{1.00} \pm \mathbf{0.91}$ | $0.42 \pm 0.58$ | $1.89 \pm 0.11$ | $\mathbf{2.10} \pm \mathbf{0.17}$ | $1.66 \pm 0.29$ | $\mathbf{1.77} \pm \mathbf{0.35}$ |
| | HUMANOID-RUN | $\mathbf{1.00} \pm \mathbf{0.88}$ | $0.66 \pm 0.87$ | $\mathbf{1.97} \pm \mathbf{0.35}$ | $1.78 \pm 1.02$ | $1.92 \pm 0.15$ | $\mathbf{2.18} \pm \mathbf{0.45}$ |
| | HUMANOID-STAND | $1.00 \pm 0.93$ | $\mathbf{1.86} \pm \mathbf{0.39}$ | $1.63 \pm 0.27$ | $\mathbf{2.01} \pm \mathbf{0.08}$ | $0.91 \pm 0.60$ | $\mathbf{1.56} \pm \mathbf{0.42}$ |
| | DOG-TROT | $1.00 \pm 0.20$ | $\mathbf{1.21} \pm \mathbf{0.37}$ | $0.91 \pm 0.35$ | $\mathbf{1.13} \pm \mathbf{0.11}$ | $0.55 \pm 0.11$ | $\mathbf{0.86} \pm \mathbf{0.43}$ |
| | DOG-WALK | $1.00 \pm 0.15$ | $\mathbf{1.16} \pm \mathbf{0.07}$ | $0.95 \pm 0.19$ | $\mathbf{1.08} \pm \mathbf{0.18}$ | $0.40 \pm 0.18$ | $\mathbf{0.87} \pm \mathbf{0.12}$ |
| | DOG-STAND | $\mathbf{1.00} \pm \mathbf{0.03}$ | $0.97 \pm 0.10$ | $0.95 \pm 0.07$ | $\mathbf{1.03} \pm \mathbf{0.03}$ | $0.90 \pm 0.05$ | $\mathbf{0.97} \pm \mathbf{0.06}$ |
| | DOG-RUN | $1.00 \pm 0.16$ | $\mathbf{1.11} \pm \mathbf{0.26}$ | $0.98 \pm 0.12$ | $\mathbf{1.29} \pm \mathbf{0.17}$ | $0.78 \pm 0.18$ | $\mathbf{1.05} \pm \mathbf{0.13}$ |
| | MEAN | 1.00 (0.88, 1.12) | **1.06** (0.93, 1.18) | 1.22 (1.11, 1.34) | **1.34** (1.21, 1.47) | 1.02 (0.89, 1.15) | **1.24** (1.12, 1.37) |
| | IQM | 1.01 (0.96, 1.08) | **1.09** (1.02, 1.14) | 1.11 (1.01, 1.27) | **1.20** (1.11, 1.40) | 0.97 (0.85, 1.10) | **1.13** (1.05, 1.24) |

**TD7 Baseline**

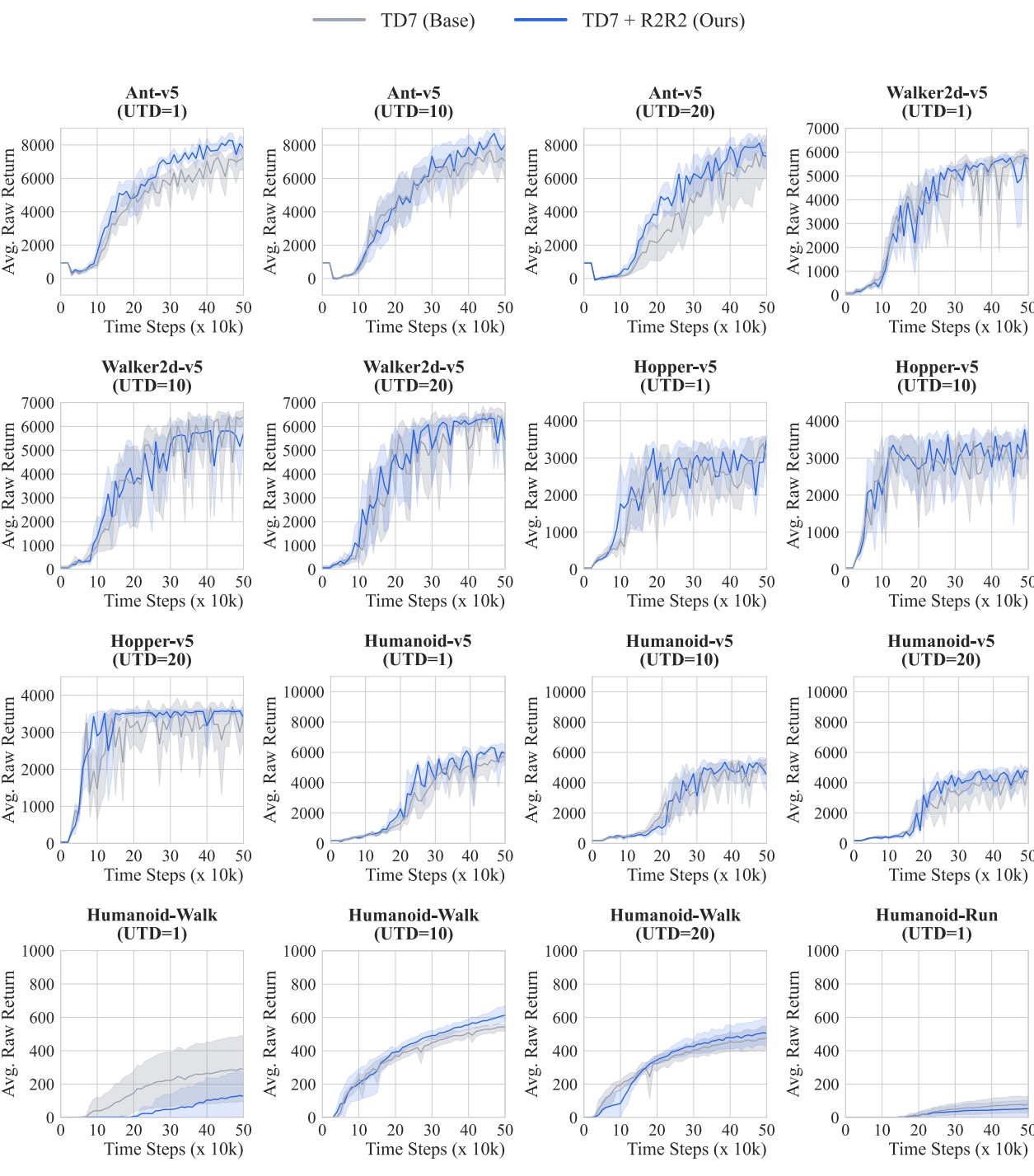

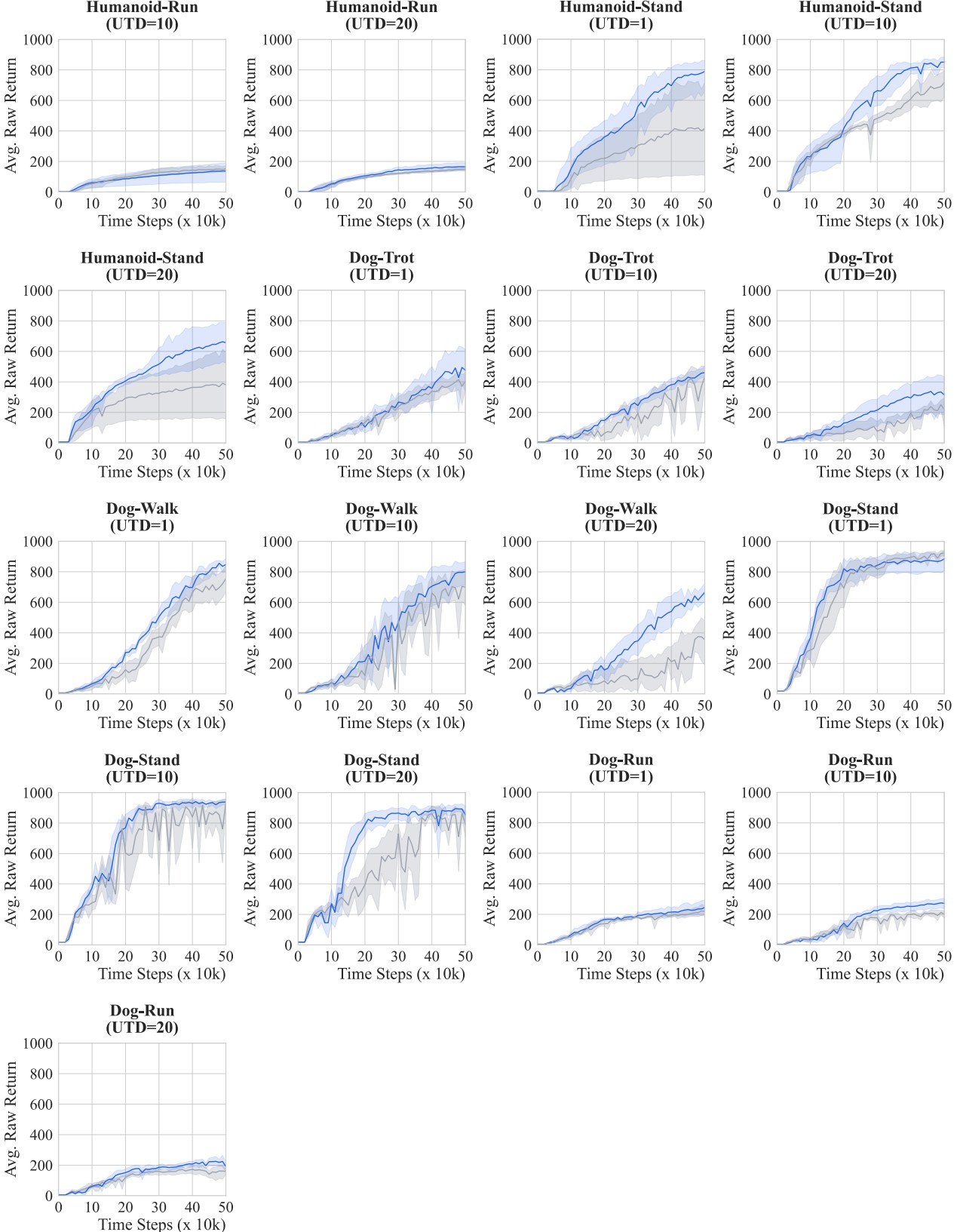

*Figure 8.* Learning curves for TD7 baseline.

## E.2. Minimalist $\phi$ baseline

*Table 8.* Raw score performance comparison on Minimalist $\phi$ (Base vs. Ours) at UTD = 1, UTD = 10, and UTD = 20. We report mean $\pm$ std.

| | ENVIRONMENT | UTD = 1 | | UTD = 10 | | UTD = 20 | |
| | | MINIMALIST $\phi$ (BASE) | MINIMALIST $\phi$ (OURS) | MINIMALIST $\phi$ (BASE) | MINIMALIST $\phi$ (OURS) | MINIMALIST $\phi$ (BASE) | MINIMALIST $\phi$ (OURS) |
|---|---|---|---|---|---|---|---|
| GYM MuJoCo | ANT-V5 | 4033.0 ± 1031.8 | **4739.5 ± 1083.8** | -415.1 ± 482.4 | **98.1 ± 162.5** | -88.5 ± 101.2 | **-41.4 ± 54.4** |
| | WALKER2D-V5 | 3645.7 ± 283.4 | **3851.2 ± 370.6** | **3237.5 ± 473.8** | 2927.8 ± 912.3 | 1729.9 ± 804.1 | **3429.7 ± 816.5** |
| | HOPPER-V5 | 2288.8 ± 754.6 | **2859.1 ± 123.2** | **2312.6 ± 866.0** | 2128.7 ± 282.1 | **2381.0 ± 418.7** | 1947.4 ± 135.5 |
| | HUMANOID-V5 | **2975.6 ± 715.1** | 1596.0 ± 456.5 | 559.5 ± 150.7 | **1164.6 ± 948.7** | 430.5 ± 173.5 | **1428.3 ± 951.6** |
| DMC-HARD | HUMANOID-WALK | **93.0 ± 203.7** | 84.2 ± 182.1 | 140.7 ± 135.6 | **220.1 ± 45.7** | 44.5 ± 96.1 | **210.2 ± 27.4** |
| | HUMANOID-RUN | 1.2 ± 0.2 | **1.5 ± 0.2** | 45.3 ± 41.2 | **71.1 ± 7.9** | 58.0 ± 33.4 | **66.9 ± 11.6** |
| | HUMANOID-STAND | 7.1 ± 1.1 | **144.9 ± 305.2** | 129.3 ± 152.1 | **266.8 ± 291.4** | 4.9 ± 0.5 | **5.2 ± 0.8** |
| | DOG-TROT | **10.2 ± 1.5** | 9.1 ± 1.6 | **10.9 ± 0.6** | 10.7 ± 0.2 | 11.0 ± 1.1 | **12.7 ± 1.1** |
| | DOG-WALK | **12.6 ± 2.1** | 9.7 ± 0.8 | **18.1 ± 7.3** | 15.0 ± 2.1 | **15.8 ± 3.6** | 14.3 ± 2.0 |
| | DOG-STAND | 32.5 ± 7.3 | **34.3 ± 6.4** | **42.5 ± 6.7** | 40.9 ± 3.6 | **47.3 ± 10.4** | 37.8 ± 2.0 |
| | DOG-RUN | **8.6 ± 1.7** | 8.1 ± 1.2 | 8.8 ± 1.3 | **8.9 ± 1.0** | 9.8 ± 1.1 | **10.3 ± 1.3** |

*Table 9.* Performance comparison on Minimalist $\phi$ (Base vs. Ours) at UTD = 1, UTD = 10, and UTD = 20. We report normalized mean $\pm$ std. The final rows report the aggregate Mean and IQM with 95% CIs.

| | ENVIRONMENT | UTD = 1 | | UTD = 10 | | UTD = 20 | |
| | | MINIMALIST $\phi$ (BASE) | MINIMALIST $\phi$ (OURS) | MINIMALIST $\phi$ (BASE) | MINIMALIST $\phi$ (OURS) | MINIMALIST $\phi$ (BASE) | MINIMALIST $\phi$ (OURS) |
|---|---|---|---|---|---|---|---|
| GYM MuJoCo | ANT-V5 | 1.00 ± 0.26 | **1.18 ± 0.27** | -0.10 ± 0.12 | **0.02 ± 0.04** | -0.02 ± 0.03 | **-0.01 ± 0.01** |
| | WALKER2D-V5 | 1.00 ± 0.08 | **1.06 ± 0.10** | **0.89 ± 0.13** | 0.80 ± 0.25 | 0.47 ± 0.22 | **0.94 ± 0.22** |
| | HOPPER-V5 | 1.00 ± 0.33 | **1.25 ± 0.05** | 1.01 ± 0.38 | 0.93 ± 0.12 | **1.04 ± 0.18** | 0.85 ± 0.06 |
| | HUMANOID-V5 | **1.00 ± 0.24** | 0.54 ± 0.15 | 0.19 ± 0.05 | **0.39 ± 0.32** | 0.15 ± 0.06 | **0.48 ± 0.32** |
| DMC-HARD | HUMANOID-WALK | **1.00 ± 2.19** | 0.91 ± 1.96 | 1.51 ± 1.46 | **2.37 ± 0.49** | 0.48 ± 1.03 | **2.26 ± 0.30** |
| | HUMANOID-RUN | 1.00 ± 0.20 | **1.27 ± 0.13** | 39.35 ± 35.85 | **61.83 ± 6.88** | 50.38 ± 29.00 | **58.16 ± 10.09** |
| | HUMANOID-STAND | 1.00 ± 0.16 | **20.27 ± 42.69** | 18.09 ± 21.27 | **37.32 ± 40.76** | 0.69 ± 0.07 | **0.73 ± 0.11** |
| | DOG-TROT | **1.00 ± 0.14** | 0.90 ± 0.16 | **1.07 ± 0.06** | 1.05 ± 0.02 | 1.08 ± 0.11 | **1.24 ± 0.11** |
| | DOG-WALK | **1.00 ± 0.17** | 0.77 ± 0.06 | **1.43 ± 0.58** | 1.19 ± 0.16 | **1.26 ± 0.28** | 1.14 ± 0.16 |
| | DOG-STAND | 1.00 ± 0.22 | **1.06 ± 0.20** | **1.31 ± 0.21** | 1.26 ± 0.11 | **1.46 ± 0.32** | 1.16 ± 0.06 |
| | DOG-RUN | **1.00 ± 0.20** | 0.94 ± 0.14 | 1.03 ± 0.15 | **1.04 ± 0.12** | 1.15 ± 0.13 | **1.21 ± 0.15** |
| | MEAN | 1.00 (0.87, 1.18) | **2.74** (0.89, 6.25) | 5.98 (2.08, 10.47) | **9.84** (4.16, 16.23) | 5.28 (1.75, 9.98) | **6.20** (2.03, 11.04) |
| | IQM | 0.98 (0.91, 1.04) | **1.01** (0.92, 1.10) | 1.03 (0.83, 1.22) | **1.08** (0.93, 2.44) | 0.87 (0.65, 1.05) | **1.05** (0.93, 1.21) |

**Minimalist $\phi$ Baseline**

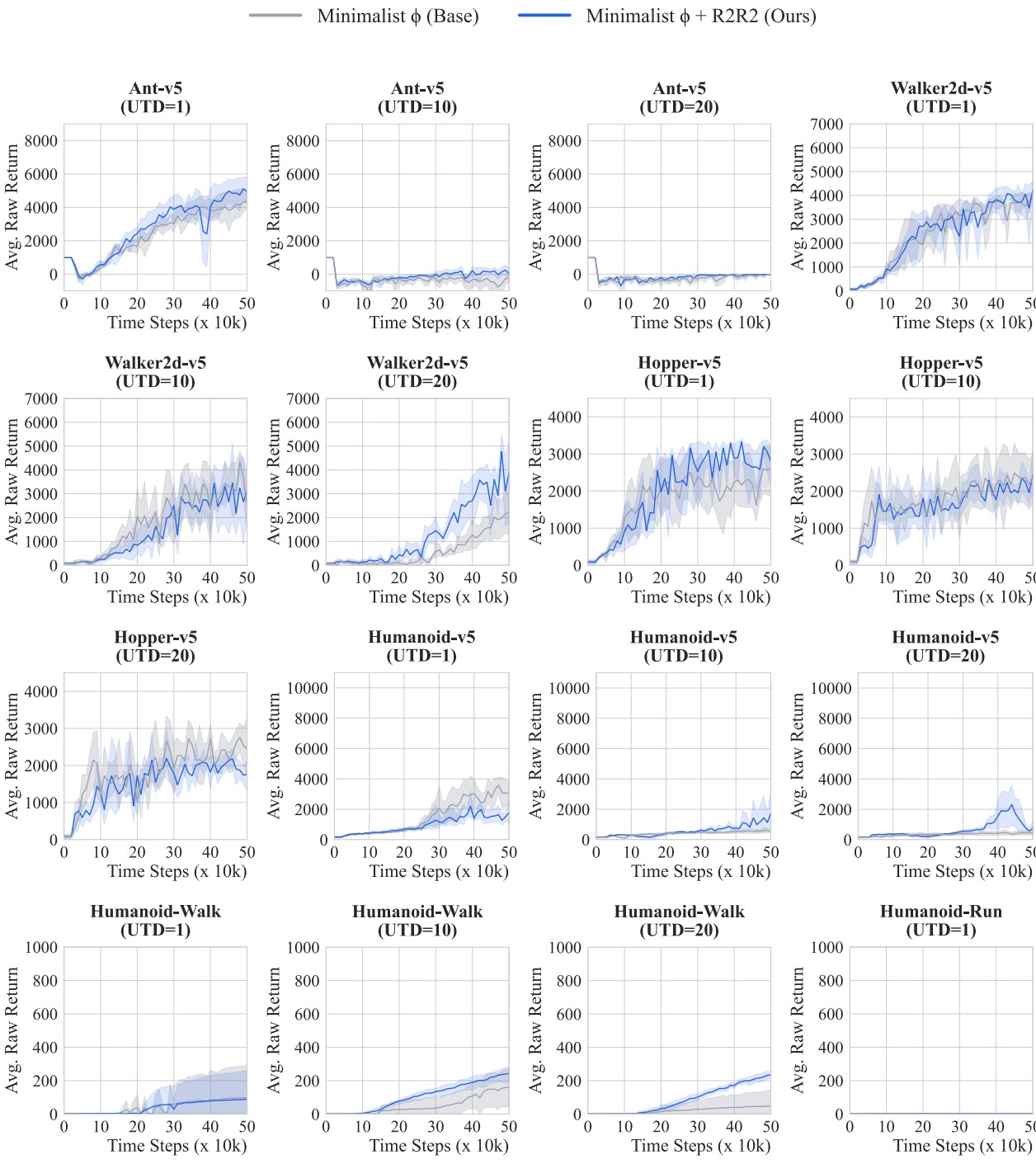

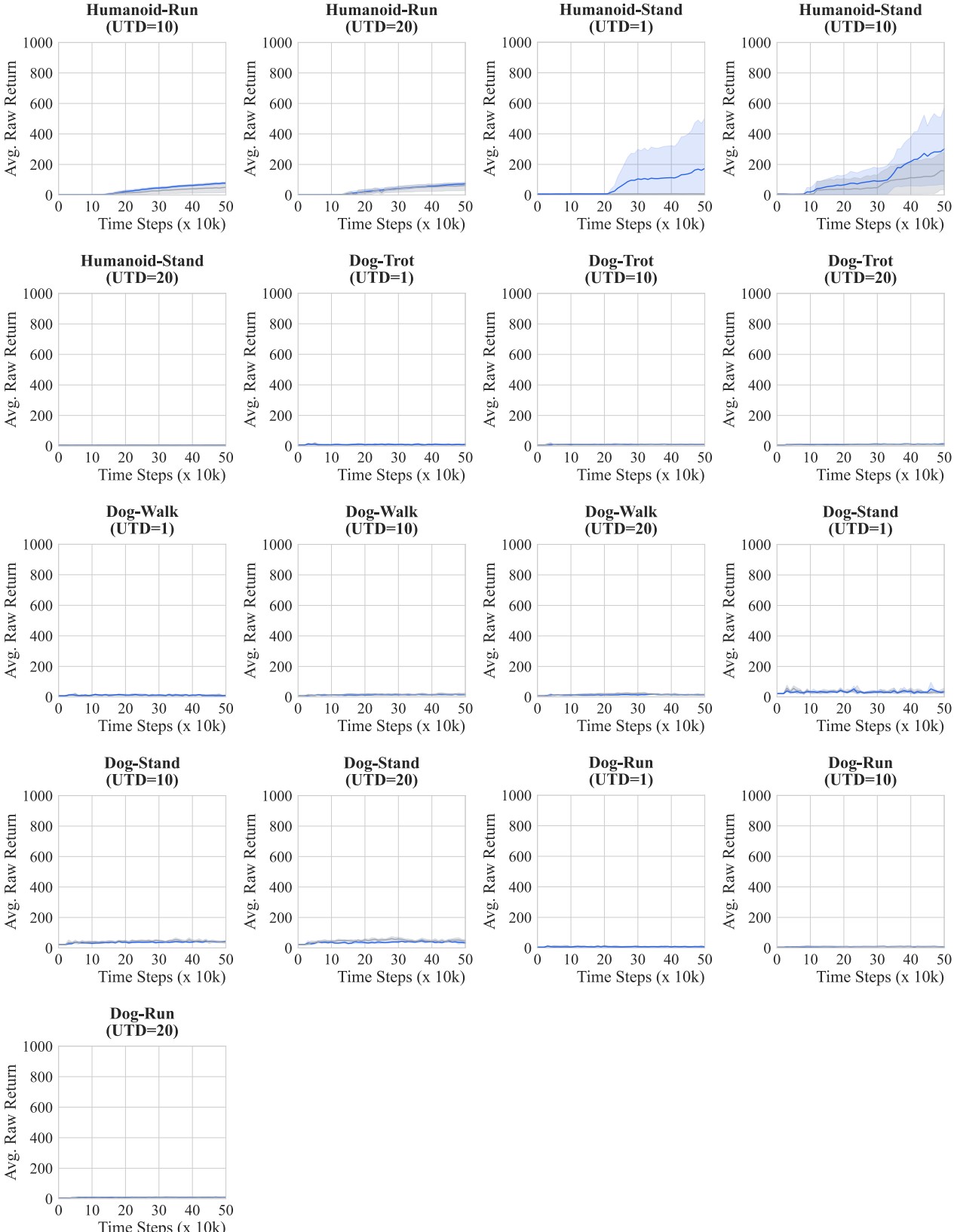

*Figure 9.* Learning curves for Minimalist $\phi$ baseline.

### E.3. TD7+Layer Normalization (LN) baseline

*Table 10.* Raw score performance comparison on TD7+LayerNorm (Fujimoto et al., 2023; Ba et al., 2016) (Base vs. Ours) at UTD = 1, UTD = 10, and UTD = 20. We report mean ± std.

| | ENVIRONMENT | UTD = 1 | | UTD = 10 | | UTD = 20 | |
| | | TD7+LN (BASE) | TD7+LN (OURS) | TD7+LN (BASE) | TD7+LN (OURS) | TD7+LN (BASE) | TD7+LN (OURS) |
|---|---|---|---|---|---|---|---|
| GYM MUJOCO | ANT-V5 | 7281.1 ± 418.2 | **7638.2 ± 421.5** | 5860.4 ± 1471.3 | **7707.6 ± 433.4** | 4130.1 ± 344.6 | **7418.2 ± 920.8** |
| | WALKER2D-V5 | **5852.7 ± 425.7** | 5625.2 ± 442.8 | 5963.6 ± 597.0 | **6040.8 ± 383.4** | 5651.7 ± 752.0 | **6421.1 ± 424.4** |
| | HOPPER-V5 | 2710.1 ± 756.1 | **2722.0 ± 483.6** | 2955.1 ± 403.8 | **3123.6 ± 675.3** | 3461.5 ± 225.3 | **3598.4 ± 56.9** |
| | HUMANOID-V5 | **5763.7 ± 562.4** | 5185.9 ± 522.2 | **5162.2 ± 335.9** | 2330.6 ± 931.0 | **4467.6 ± 543.2** | 4460.6 ± 547.5 |
| DMC-HARD | HUMANOID-WALK | 344.5 ± 206.3 | **397.1 ± 229.1** | 451.0 ± 51.0 | **553.8 ± 89.4** | 377.4 ± 53.8 | **493.4 ± 50.0** |
| | HUMANOID-RUN | 109.6 ± 62.4 | **134.9 ± 28.4** | 122.2 ± 8.2 | **170.4 ± 13.3** | 112.2 ± 13.9 | **178.3 ± 38.2** |
| | HUMANOID-STAND | 525.0 ± 88.4 | **611.4 ± 222.5** | 464.1 ± 86.5 | **657.4 ± 67.8** | 305.2 ± 171.3 | **600.2 ± 190.5** |
| | DOG-TROT | 336.1 ± 79.1 | **425.5 ± 101.0** | 327.6 ± 47.2 | **346.5 ± 63.6** | 277.7 ± 39.4 | **289.8 ± 58.1** |
| | DOG-WALK | 596.6 ± 107.9 | **701.7 ± 80.1** | 512.0 ± 125.8 | **777.3 ± 47.1** | 444.3 ± 188.8 | **560.5 ± 135.5** |
| | DOG-STAND | **927.0 ± 43.4** | 899.7 ± 41.6 | 885.0 ± 24.6 | **900.2 ± 37.1** | 782.2 ± 96.8 | **886.0 ± 20.0** |
| | DOG-RUN | 215.0 ± 23.1 | **225.7 ± 33.2** | 232.6 ± 2.1 | **244.6 ± 14.0** | 199.4 ± 17.7 | **208.1 ± 20.8** |

*Table 11.* Performance comparison on TD7 + LayerNorm (Base vs. Ours) at UTD = 1, UTD = 10, and UTD = 20. We report normalized mean ± std. The final rows report the aggregate Mean and IQM with 95% CIs.

| | ENVIRONMENT | UTD = 1 | | UTD = 10 | | UTD = 20 | |
| | | TD7+LN (BASE) | TD7+LN (OURS) | TD7+LN (BASE) | TD7+LN (OURS) | TD7+LN (BASE) | TD7+LN (OURS) |
|---|---|---|---|---|---|---|---|
| GYM MUJOCO | ANT-V5 | 1.00 ± 0.06 | **1.05 ± 0.06** | 0.81 ± 0.20 | **1.06 ± 0.06** | 0.57 ± 0.05 | **1.02 ± 0.13** |
| | WALKER2D-V5 | **1.00 ± 0.07** | 0.96 ± 0.08 | 1.02 ± 0.10 | **1.03 ± 0.07** | 0.97 ± 0.13 | **1.10 ± 0.07** |
| | HOPPER-V5 | **1.00 ± 0.28** | **1.00 ± 0.18** | 1.09 ± 0.15 | **1.15 ± 0.25** | 1.28 ± 0.08 | **1.33 ± 0.02** |
| | HUMANOID-V5 | **1.00 ± 0.10** | 0.90 ± 0.09 | **0.90 ± 0.06** | 0.40 ± 0.16 | **0.78 ± 0.09** | 0.77 ± 0.10 |
| DMC-HARD | HUMANOID-WALK | 1.00 ± 0.60 | **1.15 ± 0.67** | 1.31 ± 0.15 | **1.61 ± 0.26** | 1.10 ± 0.16 | **1.43 ± 0.15** |
| | HUMANOID-RUN | 1.00 ± 0.57 | **1.23 ± 0.26** | 1.12 ± 0.08 | **1.55 ± 0.12** | 1.02 ± 0.13 | **1.63 ± 0.35** |
| | HUMANOID-STAND | 1.00 ± 0.17 | **1.16 ± 0.42** | 0.88 ± 0.17 | **1.25 ± 0.13** | 0.58 ± 0.33 | **1.14 ± 0.36** |
| | DOG-TROT | 1.00 ± 0.24 | **1.27 ± 0.30** | 0.98 ± 0.14 | **1.03 ± 0.19** | 0.83 ± 0.12 | **0.86 ± 0.17** |
| | DOG-WALK | 1.00 ± 0.18 | **1.18 ± 0.13** | 0.86 ± 0.21 | **1.30 ± 0.08** | 0.75 ± 0.32 | **0.94 ± 0.23** |
| | DOG-STAND | **1.00 ± 0.05** | 0.97 ± 0.05 | 0.96 ± 0.03 | **0.97 ± 0.04** | 0.84 ± 0.10 | **0.96 ± 0.02** |
| | DOG-RUN | 1.00 ± 0.11 | **1.05 ± 0.15** | 1.08 ± 0.01 | **1.14 ± 0.07** | 0.93 ± 0.08 | **0.97 ± 0.10** |
| | MEAN | 1.00 (0.93, 1.07) | **1.08** (1.01, 1.16) | 1.00 (0.95, 1.05) | **1.14** (1.05, 1.23) | 0.88 (0.80, 0.94) | **1.10** (1.03, 1.19) |
| | IQM | 1.01 (0.97, 1.06) | **1.07** (1.00, 1.13) | 1.00 (0.96, 1.05) | **1.15** (1.07, 1.23) | 0.87 (0.80, 0.94) | **1.06** (0.98, 1.15) |

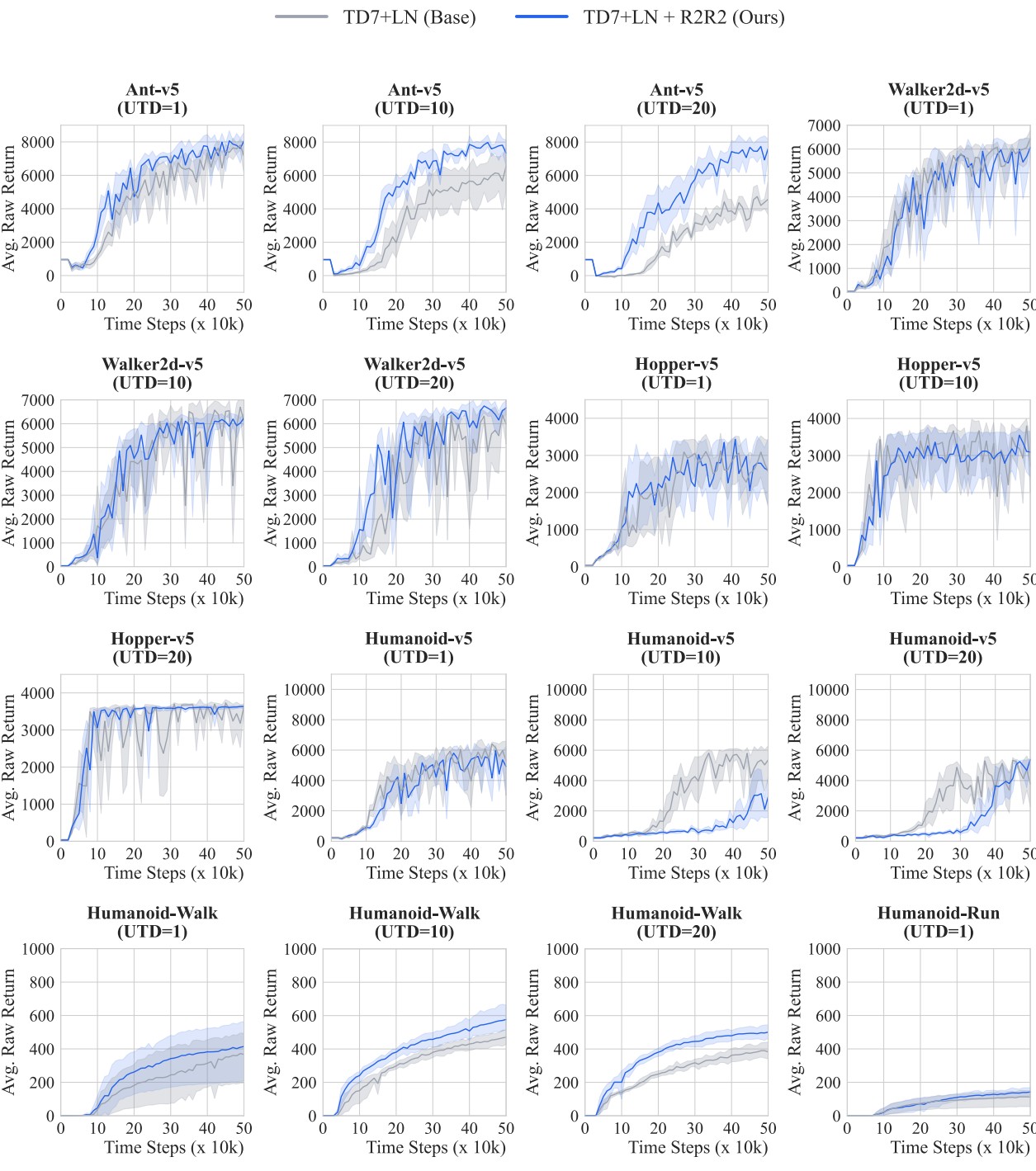

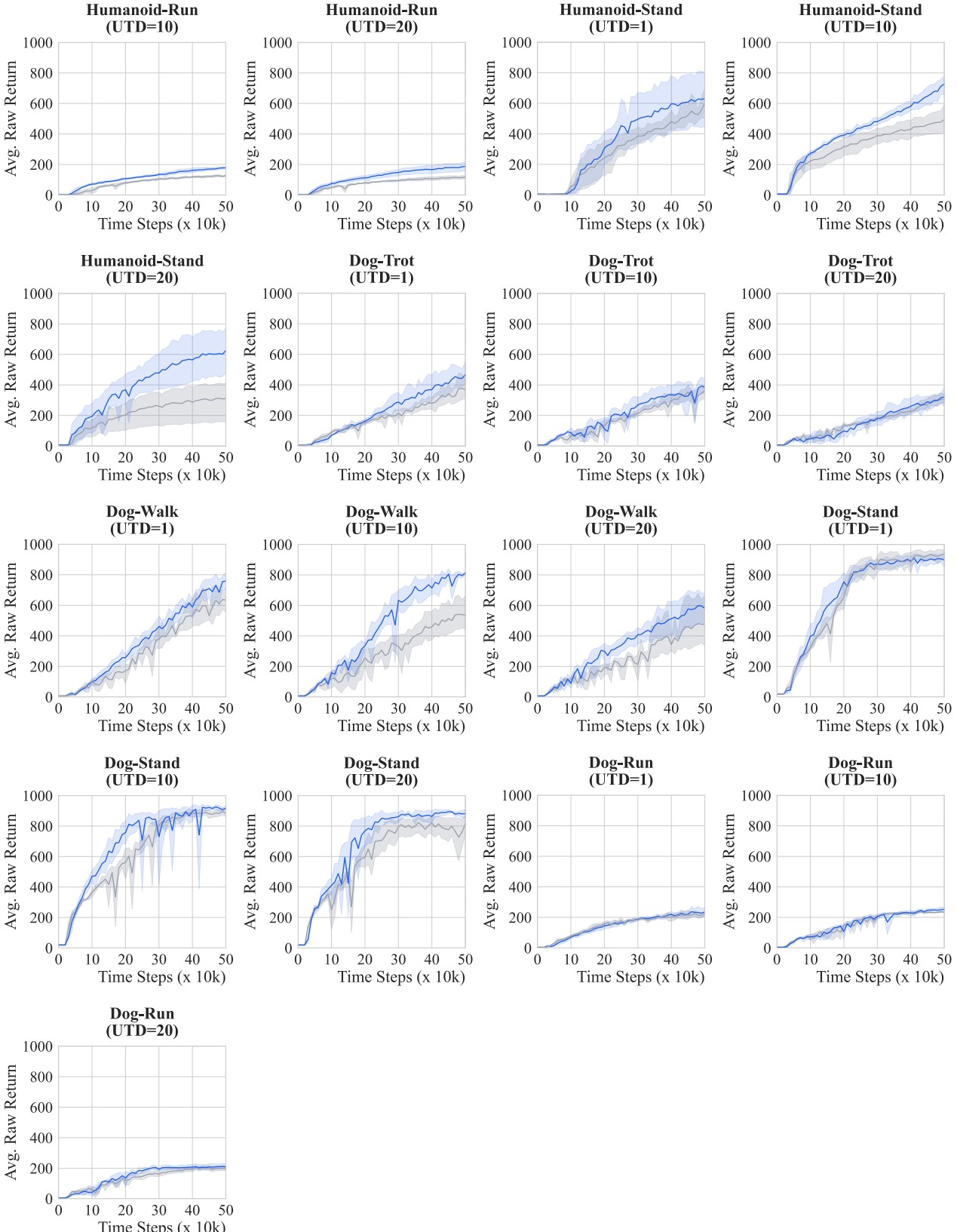

*Figure 10.* Learning curves for TD7+LN baseline.

## E.4. SimbaV2 baseline

*Table 12.* Raw score performance comparison on SimbaV2 (Lee et al., 2025b), SimbaV2-SPL, and +R2R2 at UTD = 1, UTD = 10, and UTD = 20. We report mean ± std.

| | Environment | UTD = 1 | | | UTD = 10 | | | UTD = 20 | | |
| --- | --- | --- | --- | --- | --- | --- | --- | --- | --- | --- |
| | | SimbaV2 (Base) | SimbaV2-SPL (Ours) | +R2R2 (Ours) | SimbaV2 (Base) | SimbaV2-SPL (Ours) | +R2R2 (Ours) | SimbaV2 (Base) | SimbaV2-SPL (Ours) | +R2R2 (Ours) |
| Gym MuJoCo | Ant-v5 | 5572.5 ± 1212.2 | **6953.3 ± 277.7** | 6705.1 ± 558.0 | 4861.8 ± 1800.4 | 7063.1 ± 450.4 | **7101.3 ± 349.6** | 5082.6 ± 848.1 | **7175.4 ± 293.3** | 7083.0 ± 340.0 |
| | Walker2d-v5 | **6221.2 ± 560.2** | 6036.3 ± 442.0 | 5459.0 ± 540.7 | **5884.4 ± 365.0** | 5642.7 ± 499.3 | 5823.1 ± 376.5 | **5916.9 ± 330.1** | 5715.7 ± 514.0 | 5479.6 ± 922.7 |
| | Hopper-v5 | **3437.2 ± 245.1** | 3115.8 ± 396.0 | 3308.0 ± 226.1 | 3517.0 ± 525.4 | 3534.1 ± 303.7 | **3573.1 ± 167.2** | 3614.0 ± 229.4 | **3663.2 ± 283.9** | 3527.2 ± 256.6 |
| | Humanoid-v5 | **7824.2 ± 187.2** | 7816.3 ± 839.3 | 7220.1 ± 529.9 | 9016.5 ± 550.5 | 8616.6 ± 957.9 | **9361.2 ± 372.0** | 8688.9 ± 727.6 | 8925.9 ± 318.9 | **8940.5 ± 344.2** |
| DMC-Hard | Humanoid-Walk | 478.6 ± 98.2 | 732.8 ± 92.2 | **750.1 ± 123.9** | 863.1 ± 92.2 | 890.8 ± 95.2 | **895.2 ± 68.9** | 769.6 ± 122.0 | 928.4 ± 10.4 | **935.9 ± 9.2** |
| | Humanoid-Run | 130.5 ± 24.6 | **199.4 ± 29.4** | 196.5 ± 16.3 | 259.0 ± 25.2 | **426.0 ± 42.3** | 391.5 ± 92.5 | 269.5 ± 89.0 | 389.1 ± 93.4 | **468.2 ± 44.4** |
| | Humanoid-Stand | 638.4 ± 341.5 | 848.9 ± 124.0 | **908.1 ± 38.6** | 928.3 ± 22.1 | **951.5 ± 6.3** | 944.9 ± 8.7 | 930.4 ± 6.8 | 948.8 ± 7.6 | **950.6 ± 4.8** |
| | Dog-Trot | 800.3 ± 63.8 | 867.7 ± 34.3 | **884.8 ± 36.4** | 889.7 ± 18.4 | **903.4 ± 13.9** | 849.4 ± 111.5 | **851.3 ± 75.1** | 776.3 ± 64.9 | 745.4 ± 225.9 |
| | Dog-Walk | 923.9 ± 9.8 | 924.7 ± 8.2 | **926.6 ± 6.5** | 928.6 ± 8.2 | **950.1 ± 5.4** | 942.1 ± 3.8 | 937.5 ± 6.0 | **938.3 ± 4.3** | 930.2 ± 8.3 |
| | Dog-Stand | 976.2 ± 2.2 | **980.9 ± 4.2** | 973.5 ± 5.3 | 961.3 ± 19.0 | **976.9 ± 5.3** | 956.4 ± 14.5 | 956.3 ± 12.1 | **972.3 ± 9.6** | 953.2 ± 9.1 |
| | Dog-Run | 503.5 ± 88.4 | **559.5 ± 28.0** | 538.1 ± 68.1 | 541.1 ± 126.0 | **579.1 ± 77.6** | 491.7 ± 79.2 | **484.5 ± 84.7** | 451.3 ± 53.8 | 445.0 ± 68.4 |

*Table 13.* Performance comparison on SimbaV2 (Lee et al., 2025b), SimbaV2-SPL, and +R2R2 at UTD = 1, UTD = 10, and UTD = 20. We report normalized mean ± std. The final rows report the aggregate Mean and IQM with 95% CIs.

| | Environment | UTD = 1 | | | UTD = 10 | | | UTD = 20 | | |
| --- | --- | --- | --- | --- | --- | --- | --- | --- | --- | --- |
| | | SimbaV2 (Base) | SimbaV2-SPL (Ours) | +R2R2 (Ours) | SimbaV2 (Base) | SimbaV2-SPL (Ours) | +R2R2 (Ours) | SimbaV2 (Base) | SimbaV2-SPL (Ours) | +R2R2 (Ours) |
| Gym MuJoCo | Ant-v5 | 1.00 ± 0.22 | **1.25 ± 0.05** | 1.20 ± 0.10 | 0.87 ± 0.32 | **1.27 ± 0.08** | 1.27 ± 0.06 | 0.91 ± 0.15 | **1.29 ± 0.05** | 1.27 ± 0.06 |
| | Walker2d-v5 | **1.00 ± 0.09** | 0.97 ± 0.07 | 0.88 ± 0.09 | **0.95 ± 0.06** | 0.91 ± 0.08 | 0.94 ± 0.06 | **0.95 ± 0.05** | 0.92 ± 0.08 | 0.88 ± 0.15 |
| | Hopper-v5 | **1.00 ± 0.07** | 0.91 ± 0.12 | 0.96 ± 0.07 | 1.02 ± 0.15 | 1.03 ± 0.09 | **1.04 ± 0.05** | 1.05 ± 0.07 | **1.07 ± 0.08** | 1.03 ± 0.08 |
| | Humanoid-v5 | **1.00 ± 0.02** | 1.00 ± 0.11 | 0.92 ± 0.07 | 1.15 ± 0.07 | 1.10 ± 0.12 | **1.20 ± 0.05** | 1.11 ± 0.09 | 1.14 ± 0.04 | 1.14 ± 0.04 |
| DMC-Hard | Humanoid-Walk | 1.00 ± 0.21 | 1.53 ± 0.19 | **1.57 ± 0.26** | 1.80 ± 0.19 | 1.86 ± 0.20 | **1.87 ± 0.14** | 1.61 ± 0.26 | 1.94 ± 0.02 | **1.96 ± 0.02** |
| | Humanoid-Run | 1.00 ± 0.19 | **1.53 ± 0.23** | 1.51 ± 0.13 | 1.99 ± 0.19 | **3.26 ± 0.32** | 3.00 ± 0.71 | 2.07 ± 0.68 | 2.98 ± 0.72 | **3.59 ± 0.34** |
| | Humanoid-Stand | 1.00 ± 0.54 | 1.33 ± 0.19 | **1.42 ± 0.06** | 1.45 ± 0.04 | **1.49 ± 0.01** | 1.48 ± 0.01 | 1.46 ± 0.01 | **1.49 ± 0.01** | 1.49 ± 0.01 |
| | Dog-Trot | 1.00 ± 0.08 | 1.08 ± 0.04 | **1.11 ± 0.05** | 1.11 ± 0.02 | **1.13 ± 0.02** | 1.06 ± 0.14 | **1.06 ± 0.09** | 0.97 ± 0.08 | 0.93 ± 0.28 |
| | Dog-Walk | **1.00 ± 0.01** | **1.00 ± 0.01** | **1.00 ± 0.01** | 1.01 ± 0.01 | **1.03 ± 0.01** | 1.02 ± 0.00 | **1.02 ± 0.01** | 1.02 ± 0.01 | 1.01 ± 0.01 |
| | Dog-Stand | 1.00 ± 0.00 | **1.01 ± 0.00** | 1.00 ± 0.01 | 0.99 ± 0.02 | **1.00 ± 0.01** | 0.98 ± 0.02 | 0.98 ± 0.01 | **1.00 ± 0.01** | 0.98 ± 0.01 |
| | Dog-Run | 1.00 ± 0.18 | **1.11 ± 0.06** | 1.07 ± 0.14 | 1.08 ± 0.25 | **1.15 ± 0.15** | 0.98 ± 0.16 | **0.96 ± 0.17** | 0.90 ± 0.11 | 0.88 ± 0.14 |
| | Mean | 1.00 (0.95, 1.04) | **1.16** (1.09, 1.22) | 1.15 (1.08, 1.22) | 1.22 (1.12, 1.33) | **1.38** (1.22, 1.58) | 1.35 (1.20, 1.53) | 1.20 (1.10, 1.32) | 1.34 (1.18, 1.52) | **1.38** (1.19, 1.61) |
| | IQM | 1.01 (0.98, 1.04) | **1.10** (1.04, 1.18) | 1.09 (1.03, 1.18) | 1.12 (1.05, 1.23) | **1.16** (1.09, 1.29) | 1.15 (1.07, 1.27) | 1.09 (1.03, 1.19) | **1.13** (1.05, 1.26) | 1.12 (1.04, 1.26) |

## SimbaV2 Baseline

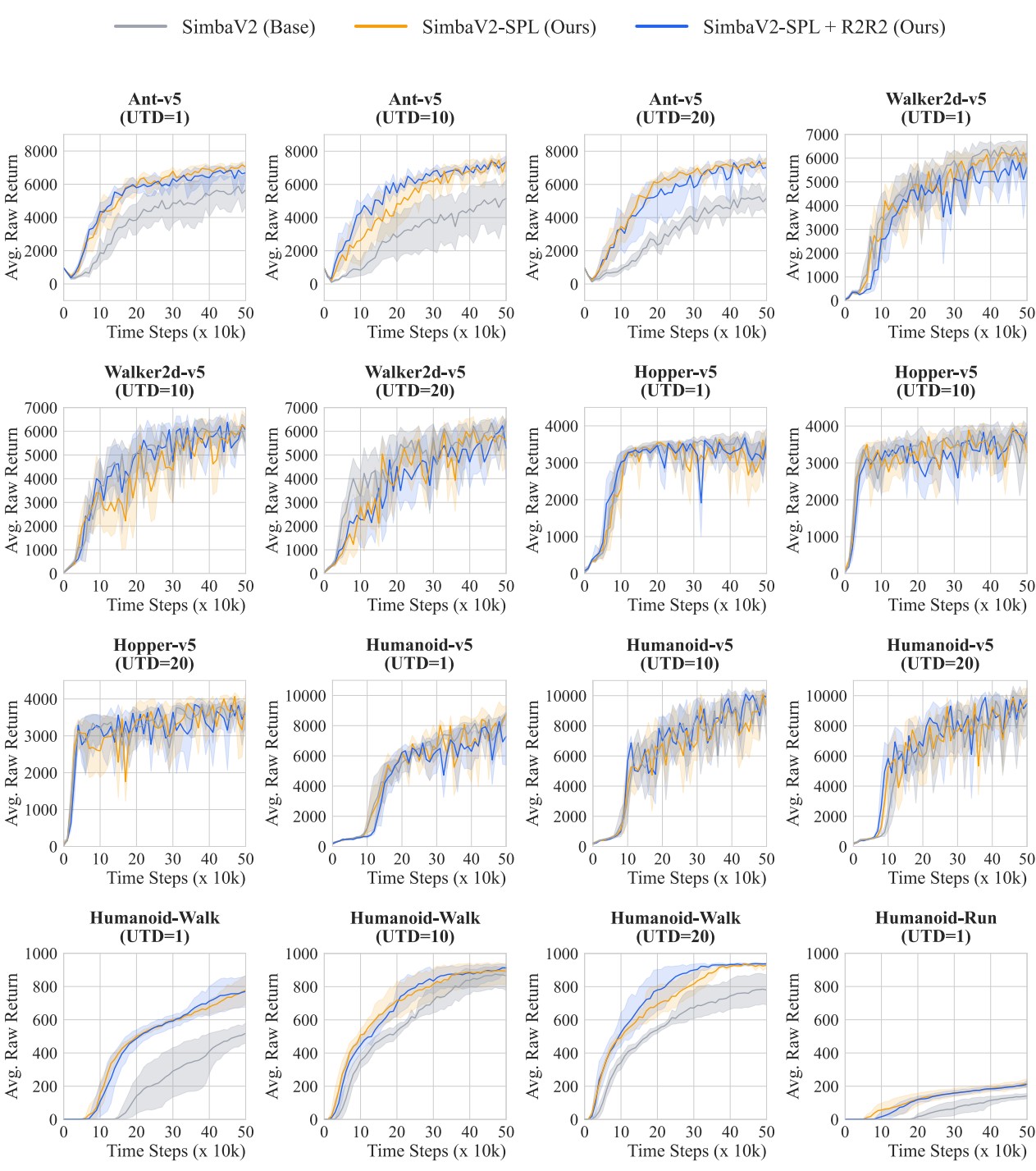

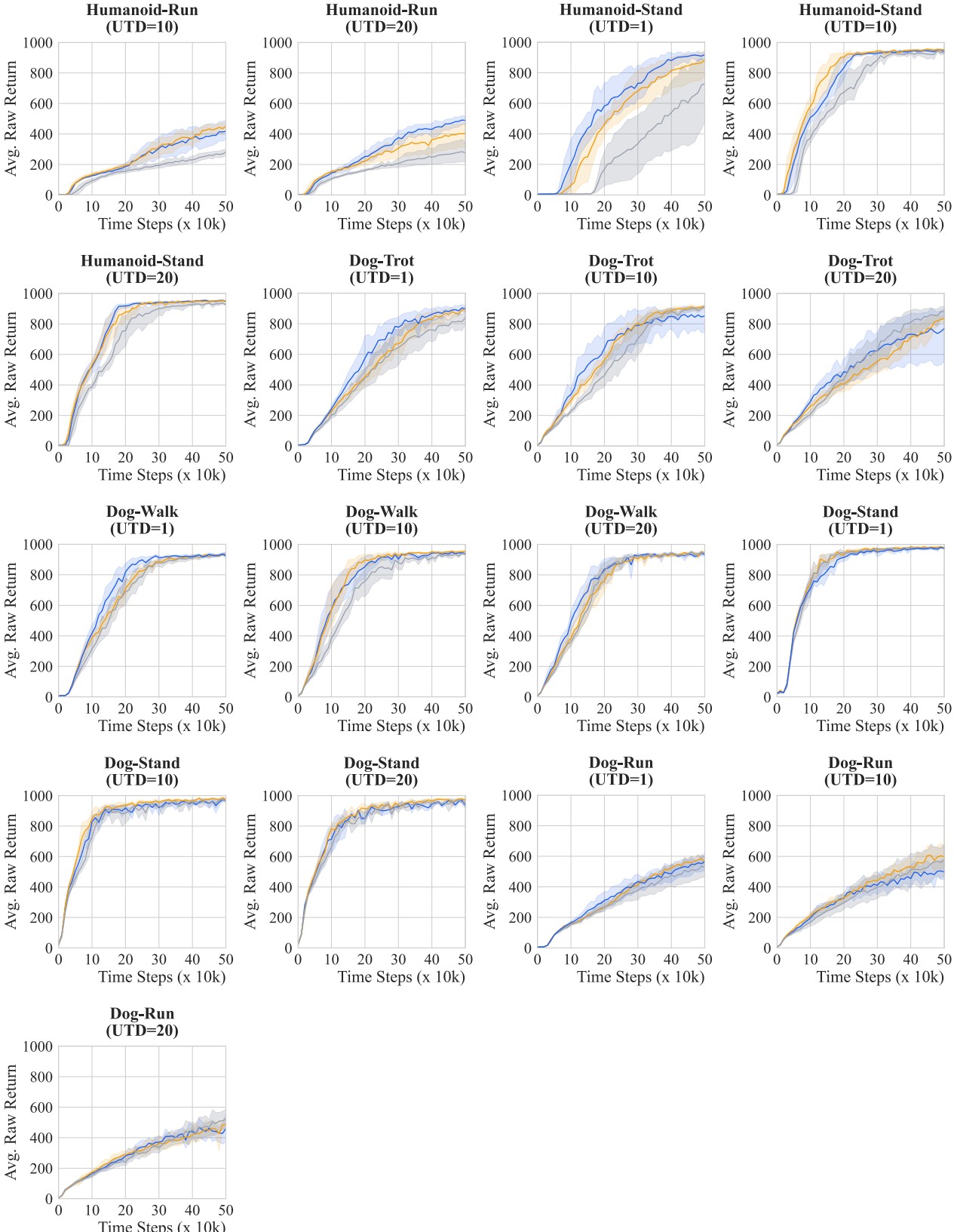

*Figure 11.* Learning curves for SimbaV2 baseline.

