# OpenReview forum: "R2R2: Robust Representation for Intensive Experience Reuse via Redundancy Reduction in Self-Predictive Learning"
_ICML.cc/2026/Conference — ICML 2026 regular_

### Official Review · Reviewer_2NvF · 2026-02-25

**Soundness:** 2
**Presentation:** 3
**Significance:** 2
**Originality:** 3
**Overall Recommendation:** 4
**Confidence:** 4

**Summary:**

This paper investigates the representation-level instability of Self-Predictive Learning (SPL) in deep reinforcement learning, particularly under high Update-to-Data (UTD) regimes. To address this instability, the authors propose Representation via Redundancy Reduction (R2R2), an approach that introduces two additional regularization losses to the standard self-prediction objective. The work provides a theoretical justification for these auxiliary losses and empirically validates R2R2 by integrating it with several existing methods across the MuJoCo and DeepMind Control (DMC) Hard benchmarks.

**Compliance With Llm Reviewing Policy:**

Affirmed.

**Final Justification:**

The authors’ rebuttal has addressed my concerns, and I therefore raise my score accordingly.

**Key Questions For Authors:**

1. Could the authors provide additional empirical comparisons against other SPL methods (e.g., MRQ and SPR-VICRegs) to better contextualize R2R2's performance?

**Limitations:**

yes

**Strengths And Weaknesses:**

**Strengths**

1. The paper offers a rigorous theoretical understanding of SPL dynamics and representation collapse within reinforcement learning.
2. The authors extensively evaluate R2R2 by combining it with four distinct baselines (TD7, TD7+LN, minimalist $\phi$, simbav2). This broad application successfully demonstrates the method's versatility under both low and high UTD regimes.
3. The inclusion of ablation studies effectively isolates and validates the individual empirical contributions of the two proposed regularization terms.

**Weaknesses**

1. The empirical evaluation critically lacks comparison with other recent and relevant SPL methods in the literature. Without comparisons to existing approaches such as MRQ [1] and SPR-VICRegs [2], it is difficult to gauge where R2R2 stands relative to previous approaches.

   [1] Scott Fujimoto et al. Towards General-Purpose Model-Free Reinforcement Learning. ICLR 2025.

   [2] Ömer Veysel Ça ̆gatan, Barı ̧s Akgün. Uncovering RL Integration in SSL Loss: Objective-Specific Implications for Data-Efficient RL. RLC 2025.

2.  Normalizing the UTD=1 baseline scores to 1.0 can be misleading. If a baseline method completely fails (achieving a near-zero absolute score) at UTD=1, a seemingly massive relative improvement multiplier upon adding R2R2 might still equate to a trivially poor absolute performance. For instance, as shown in Table 8, the minimalist $\phi$ baseline performs poorly on the DMC benchmark. Consequently, the large normalized improvements presented in Table 1 distort the actual capabilities of the agent, masking the fact that the absolute performance remains poor. I strongly suggest the authors replace the normalized metrics with the raw returns directly in Table 1 to allow for a transparent and accurate assessment of the method's efficacy.

3. The experiments are conducted using only 5 random seeds. Given the high variance and sensitivity inherent to deep RL experiments, 5 seeds are insufficient.

4. The method introduces two additional regularization losses but lacks a comprehensive sensitivity analysis for the corresponding hyperparameters ($\lambda_{RR}$ and $\lambda_{VAR}$).

---

> ### Author Rebuttal · Authors · 2026-03-31
>
> We thank you for your review. We agree with the points you raised and have addressed each of your comments below.
>
> **W1 and Q1. Could the authors provide additional empirical comparisons against other SPL methods (e.g., MR.Q and SPR-VICRegs)?**
>
> > Our method does not directly compete with existing SPL-based RL methods; rather, it is complementary. Therefore, it must be integrated with existing baselines, and comparisons should be made based on whether our method is applied or not. We have conducted additional experiments on the SPL methods you mentioned, namely SPR and MR.Q.
> >
> > Regarding the experiments on these two baselines, we would like to provide the following comments:
> >
> > **1. SPR baseline:**
> > First, regarding the suggested SPR+VICRegs [R1], as there is no publicly available code or hyperparameter information, we cited the reported results (Table 3 of the corresponding paper). As shown in the table, applying VICRegs actually degrades the baseline performance.
> >
> >||VR-L|VR-H|SPR|
> >|-|-|-|-|
> >|Pong|-6.3|-10.1|**-5.4**|
> >
> > Since the code for the SPR [R2] baseline itself is publicly available, we implemented and evaluated our method as SPR+(Ours).
> > Regarding implementation details, the SPR encoder's representation ($z$) is a spatial feature map. Directly flattening $z$ to enforce orthogonality disrupts this spatial locality, potentially degrading performance. To mitigate this, an intermediate projector after flattening may be necessary. Therefore, we evaluated R2R2 on the SPR baseline both directly and with this added projector. Results confirm that both configurations operate effectively. (The metric is calculated following the main paper.)
> >
> > [Graph](https://drive.google.com/file/d/1EK--0p7OlpLKNXbQVyfZKKYVeM-cFt6i/view?usp=drive_link)
> >
> >|| SPR | +R2R2 | +R2R2+Projector |
> >|-| :---: | :---: | :---: |
> >|Atari-Pong-V5(2,100k)| -10.0 | -8.3 |**-6.5**|
> >
> > **2. MR.Q baseline:**
> > The original implementation of MR.Q already incorporates linear layers designed to eliminate locality during the process of extracting (z) from image inputs. Thanks to this architectural choice, R2R2 can be applied directly. The experimental results demonstrate an improvement in performance. (For Humanoid-Run at UTD=20, although the computationally intensive training is still ongoing, the performance superiority is already clearly evident.)
> >
> > [Graph](https://drive.google.com/file/d/1vC9zWl7liHk-jgTbEL_pYA3GXrY1GwQW/view?usp=drive_link)
> >
> > |Env(UTD,Steps)|MRQ|+R2R2|
> > |-|-|-|
> >|Atari-Pong-V5(1,800k)|14.5|**15.4**|
> >|Humanoid-Run(1,500k)|171.0|**191.1**|
> >|Visual-Quadruped-Run(1,500k)|467.7|**505.6**|
> >|Humanoid-Run(20,290k)|62.0|**115.4**|
> >
> > [R1] Ömer Veysel Ça ̆gatan, Barı ̧s Akgün. Uncovering RL Integration in SSL Loss: Objective-Specific Implications for Data-Efficient RL. RLC 2025.
> >
> > [R2] Max Schwarzer, Ankesh Anand, Rishab Goel, R. Devon Hjelm, Aaron C. Courville, Philip Bachman: Data-Efficient Reinforcement Learning with Self-Predictive Representations. ICLR 2021
>
> **W2. Concerns Regarding Normalized Metrics and Request for Raw Returns**
>
> >We share your concern. However, normalization was necessary to aggregate heterogeneous score ranges across environments without distortion. To address this, we included the raw scores and learning curves in the Appendix. The revised main text will explicitly note this score inflation effect and direct readers to the Appendix.
>
> **W3. Five seeds are insufficient**
>
> > We acknowledge your concern regarding this point. However, we politely ask for your understanding, considering that evaluating with 5 random seeds is still a accepted practice in the RL community, as seen in recent works such as SimbaV2. Furthermore, we would appreciate it if you could take into account that our experiments focus on high UTD regimes, which are highly computationally demanding.
>
> **W4. Lacks a comprehensive sensitivity analysis for the corresponding hyperparameters ($\\lambda\_{RR}$ and $\\lambda\_{Var}$).**
>
> > We have attached the performance graphs and tables below, showing the results when scaling $\\lambda\_{RR}$ and $\\lambda\_{Var}$ up and down by a factor of 10.
> >
> > [Graph_Sweep_RR](https://drive.google.com/file/d/1G0hrlrz_0nqSgZS444cVXnRkOqa0US0p/view?usp=drive_link)
> >
> > [Graph_Sweep_Var](https://drive.google.com/file/d/1ZJBMpYbTYGm7Wr-7_HklCTZAIM6bpK6x/view?usp=drive_link)
> >
> > When measuring performance with varying coefficients, our proposed values are not necessarily the absolute best in every single environment. For instance, in the Walker2d environment, a setting of $\\lambda\_{RR}$=0.1 yields better results than our proposed 0.01. However, this same setting performs worse than our configuration in the Hopper environment. Consequently, we can confirm that our proposed values serve as a robust and reasonable choice to be used universally as a single default setting across all environments.

---

> > ### Author Rebuttal · Reviewer_2NvF · 2026-04-01
> >
> > I appreciate the authors’ thorough and detailed responses, and I have adjusted my score accordingly.

---

> > > ### Author Response · Authors · 2026-04-03
> > >
> > > Thank you very much for your constructive critique and for raising your recommendation score. Your review provided us with an excellent opportunity to verify the new baseline. Thank you again for your time and guidance.

---

### Official Review · Reviewer_n7j8 · 2026-03-08

**Soundness:** 3
**Presentation:** 2
**Significance:** 2
**Originality:** 3
**Overall Recommendation:** 4
**Confidence:** 3

**Summary:**

This paper addresses the challenge of representation instability in RL arising from high Update-to-Data ratios. The authors propose R2R2, a regularization-based approach that effectively mitigates representation-level instability under high-UTD regimes. Empirical evaluations across 11 continuous control tasks demonstrate that R2R2 yields significant performance improvements.

**Compliance With Llm Reviewing Policy:**

Affirmed.

**Final Justification:**

This paper addresses the challenge of representation instability in RL arising from high Update-to-Data ratios. I thank the authors for their detailed and convincing rebuttal, which has successfully resolved my initial concerns. Accordingly, I have raised my score.

**Key Questions For Authors:**

1. Could the authors provide additional quantitative metrics to substantiate that the observed performance improvements are directly attributable to the mitigation of potential representational degradation?

2. Section 4.2 posits that explicit redundancy reduction schemes based on VICReg (Bardes et al., 2022) is better suited than Barlow Twins (Zbontar et al., 2021) for preserving the superior characteristics of the original SPL. Is this conclusion supported by empirical evidence?

3. The proposed method introduces additional regularization terms with associated coefficients. Could the authors provide a sensitivity analysis illustrating how performance varies across a range of coefficient values?

**Limitations:**

yes

**Strengths And Weaknesses:**

Strengths:
1. Identifies and mathematically proves the conflict between zero-centering and the spectral requirements of SPL, offering a clear justification for non-centered regularization.
2. R2R2 is complementary to both value-based stabilization methods and modern architectural improvements (e.g., SimbaV2), allowing for cumulative performance gains.

Weaknesses:
1. Currently evaluated only on low-dimensional state-based continuous control tasks; effectiveness on high-dimensional pixel-based visual RL remains unverified.
2. The identification of representational degradation is proxy-based, deduced primarily from observed performance deterioration rather than other quantitative metrics.

---

> ### Author Rebuttal · Authors · 2026-03-31
>
> Thank you for your valuable review. We apologize for the lack of experimental results addressing the points you raised.
>
> **W1. Effectiveness on high-dimensional pixel-based visual RL remains unverified.**
>
> > Our method serves to enhance performance by being integrated with existing baselines. Because all the baselines we initially adopted deal exclusively with low-dimensional vector states, our validation on pixel inputs was lacking. However, as validating pixel inputs is undeniably an important aspect, we have conducted additional experiments using SPR [R1] and MR.Q [R2], which are SPL methods that take images as inputs.
> >
> > Regarding the experiments on these two baselines, we would like to provide the following comments:
> >
> > **1. SPR baseline:**
> > Regarding implementation details, the SPR encoder's representation (z) is a spatial feature map. Directly flattening to enforce orthogonality disrupts this spatial locality, potentially degrading performance. To mitigate this, an intermediate projector after flattening may be necessary. Therefore, we evaluated R2R2 on the SPR baseline both directly and with this added projector. Results confirm that both configurations operate effectively. (The metric is calculated following the main paper.)
> >
> > [Graph](https://drive.google.com/file/d/1EK--0p7OlpLKNXbQVyfZKKYVeM-cFt6i/view?usp=drive_link)
> >
> >|| SPR |+R2R2|+R2R2+Projector|
> >|-|-|-|-|
> >|Atari-Pong-V5(2,100k)|-10.0|-8.3|**-6.5**|
> >
> > **2. MR.Q baseline:**
> > The original implementation of MR.Q already incorporates linear layers to eliminate locality during the process of extracting z from image inputs. Thanks to this architecture, R2R2 can be applied directly. The experimental results demonstrate an immediate improvement in performance.
> >
> > [Graph](https://drive.google.com/file/d/1-S_QYvigOlHmu-PVbRpglf_RqB3xwOVv/view?usp=drive_link)
> >
> > |Env(UTD,Steps)|MRQ|+R2R2|
> > |-|-|-|
> >|Atari-Pong-V5(1,800k)|14.5|**15.4**|
> >|Visual-Quadruped-Run(1,500k)|467.7|**505.6**|
> >
> > Ultimately, both experiments demonstrate that even in environments receiving pixel inputs, as long as the specific property of locality is adequately controlled, our method (R2R2) effectively synergizes with existing SPL-based methods to contribute to overall performance improvements.
> >
> > [R1] Max Schwarzer, Ankesh Anand, Rishab Goel, R. Devon Hjelm, Aaron C. Courville, Philip Bachman: Data-Efficient Reinforcement Learning with Self-Predictive Representations. ICLR 2021
> >
> > [R2] Fujimoto, Scott, et al. "Towards General-Purpose Model-Free Reinforcement Learning." The Thirteenth International Conference on Learning Representations.
>
> **W2 and Q1. Other quantitative metrics related to representational degradation.**
>
> > The spectral analysis of the representation in Section 5.6 serves as direct evidence. When the UTD ratio is high, the representational dimensions are not utilized efficiently, leading to a cut-off phenomenon in the tail and a gradual decrease in the effective rank (ER) (Fig. 6 (Left)). In other words, a partial dimensional collapse occurs. To empirically demonstrate this further, the graph below compares the effective rank over training time.
> >
> > [Graph](https://drive.google.com/file/d/1Ryk5NRcswYjYV97rBbFimjS8bawFQIza/view?usp=drive_link)
> >
> > Consequently, this representational degradation is characterized by the lowered ER and the cut-off phenomenon in high-frequency components at high UTD regimes. Applying our R2R2 effectively mitigates these issues by enforcing the efficient utilization of dimensions.
>
> **Q2. Empirical Evidence Comparing VICReg and Barlow Twins.**
>
> > We have conducted a direct comparison with the approach incorporating Barlow Twins (BT), and the resulting graph and table are provided in the link below.
> >
> > [Graph](https://drive.google.com/file/d/1h7swn3kkUXzQhhjAwRm_Mr6G6cP-rDOS/view?usp=drive_link)
> >
> >|| TD7+R2R2 | TD7+R2R2 (VICReg->BT) |
> >|-|-|-|
> >|Humanoid-Run (UTD=20)|**162.5**| 40.6 |
>
> **Q3. Could the authors provide a sensitivity analysis illustrating how performance varies across a range of coefficient values?**
>
> > We have attached the performance graphs and tables below, showing the results when scaling $\\lambda\_{RR}$ and $\\lambda\_{Var}$ up and down by a factor of 10.
> >
> > [Graph_Sweep_RR](https://drive.google.com/file/d/1G0hrlrz_0nqSgZS444cVXnRkOqa0US0p/view?usp=drive_link)
> >
> > [Graph_Sweep_Var](https://drive.google.com/file/d/1ZJBMpYbTYGm7Wr-7_HklCTZAIM6bpK6x/view?usp=drive_link)
> >
> > When measuring performance with varying coefficients, our proposed values are not necessarily the absolute best in every single environment. For instance, in the Walker2d environment, a setting of $\\lambda\_{RR}$=0.1 yields better results than our proposed 0.01. However, this same setting performs worse than our configuration in the Hopper environment. Consequently, we can confirm that our proposed values serve as a robust and reasonable choice to be used universally as a single default setting across all environments.

---

> > ### Author Rebuttal · Reviewer_n7j8 · 2026-04-02
> >
> > Thanks for the detailed reply. The rebuttal resolved my concerns.

---

> > > ### Author Response · Authors · 2026-04-03
> > >
> > > Thank you for the constructive feedback and the upgraded score. Your suggestions prompted us to evaluate a new baseline, adding substantial value and persuasiveness to our research. We are very grateful for your guidance.

---

### Official Review · Reviewer_ausv · 2026-03-08

**Soundness:** 2
**Presentation:** 3
**Significance:** 2
**Originality:** 2
**Overall Recommendation:** 4
**Confidence:** 4

**Summary:**

This paper studies the combination of two prominent directions for improving sample efficiency in RL: representation learning and high update-to-data (UTD) ratios. While prior work has demonstrated that both strategies individually improve sample efficiency, the learning dynamics of representations under high-UTD regimes have received limited attention. The authors argue that existing value-centric advancements are insufficient to address representation-level instability caused by intensive experience reuse in high-UTD settings. To address this issue, the paper proposes R2R2, an improved representation learning scheme derived from modifications to VICReg tailored to Markov decision processes. The authors provide theoretical analysis to justify key design choices and empirically demonstrate the performance advantages of R2R2 across a wide range of tasks.

**Compliance With Llm Reviewing Policy:**

Affirmed.

**Final Justification:**

The authors’ rebuttal has addressed my concerns, and I therefore raise my score accordingly.

**Key Questions For Authors:**

1. What exactly is "representation-level instability"? The authors claim that R2R2 mitigates representation-level instability caused by intensive experience reuse. Does this refer to representation collapse? Or does "instability" refer to excessive drift or oscillation of representations during training? A more precise definition would be helpful.
2. Why is the auxiliary projector considered redundant? The paper states: "This design choice is grounded in our theoretical analysis, which provides a precise understanding of SPL dynamics, indicating that the auxiliary projector is redundant." Could the authors provide a more detailed explanation?
3. Critic input design in Fig. 3.  In the SimbaV2-SPL scheme, the critic input includes $s_{t}, a_{t}, Z_{t}$, which seems reasonable. Why is $Z^\prime_{t+1}$ also included?
4. Possible diagram issue in Fig. 3. In the lower-left of Fig. 3, should SPL be computed between $Z_{t+1}$and $Z^\prime_{t+1}$?

Overall, I think this paper is interesting. If the authors can address my concerns in the rebuttal, I could consider increasing my scores.

**Limitations:**

yes

**Strengths And Weaknesses:**

**Strengths**

1. Insightful problem identification. Recent work on high-UTD regimes has largely focused on plasticity loss and improving Q-value estimation. The authors identify a relatively underexplored issue: representation-level instability under high UTD. This is a meaningful and timely observation.
2. Theoretically motivated modification of VICReg. The paper builds upon VICReg and provides theoretical analysis formally demonstrating the conflict between state predictive learning (SPL) and zero-centering. The proposed adaptation to the RL setting is principled and well-motivated, leading to the R2R2 algorithm.
3. Comprehensive experimental evaluation. The authors conduct extensive experiments to analyze R2R2’s characteristics, including: performance under high UTD, compatibility with different algorithms, the effectiveness of design choices, etc.

**Weaknesses**

1. Lack of direct evidence for representation degradation. Although the paper repeatedly emphasizes representation-level degradation under high UTD, there is no explicit experimental visualization or quantitative analysis demonstrating this phenomenon.
2. Limited gains when combined with SimbaV2-SPL. The performance improvements of R2R2 appear relatively modest when combined with SimbaV2-SPL. It would be helpful for the authors to analyze the underlying reasons for this phenomenon. One possible explanation is that both R2R2 and SimbaV2 alleviate encoder plasticity loss, meaning their effects may overlap rather than being complementary. If so, the observed gains may not primarily stem from mitigating the claimed “representation-level instability,” but instead from mechanisms similar to those in SimbaV2. (In this regard, it would be valuable for the authors to further clarify what exactly is meant by representation-level instability)

---

> ### Author Rebuttal · Authors · 2026-03-31
>
> Thank you for your detailed review. We also apologize for any lack of clarity regarding our terminology in the paper.
>
> **W1 & Q1. What exactly is "representation-level instability"?**
>
> >As you pointed out, the term 'representation-level instability' was used somewhat broadly in our paper. To clarify, the instability we observed and aimed to mitigate is precisely the 'partial dimensional collapse' you mentioned.
> >
> >This collapse is empirically supported by the spectral analysis in Section 5.6. Under high-UTD conditions, we observe a sharp cut-off phenomenon in the tail indices of the baseline, which indicates a loss of capacity to capture fine-grained, high-frequency features. To further substantiate this, we are providing an additional graph that monitors the changes in the Effective Rank (ER) of the representations during training (Humanoid-Run, UTD=20, TD7 baseline). As the graph demonstrates, the baseline TD7 without R2R2 clearly suffers from partial dimensional collapse at high UTD, resulting in its ER remaining consistently lower compared to the agent trained with R2R2.
> >
> > [Graph](https://drive.google.com/file/d/1Ryk5NRcswYjYV97rBbFimjS8bawFQIza/view?usp=drive_link)
> >
> >We cautiously hypothesize that this partial dimensional collapse stems from the moving target problem inherent in Self-Predictive Learning (SPL). While SPL's bootstrapping naturally causes representation drift, this can escalate into 'excessive drift' in high-UTD regimes. Consequently, the network is forced to abandon predicting fine-grained, high-frequency details, leading to the observed collapse.
>
> **W2. Limited gains when combined with SimbaV2-SPL.**
> > We appreciate your insightful comment. In terms of purpose, SimbaV2, structural normalization (e.g., LN), and our R2R2 all share the common goal of mitigating the loss of plasticity.
> >
> > However, their mechanisms differ. While SimbaV2 directly controls the magnitudes of features and network weights, our approach specifically targets the SPL encoder to enforce representation orthogonality, thereby maximizing dimensional efficiency.
> >
> > Consequently, as shown in our responses to W1 and Q1, our method prevents partial dimensional collapse, such as the tail cut-off in the representation's spectral analysis. We attribute the marginal performance difference to a ceiling effect, as SimbaV2-SPL already yields sufficiently high performance.
> >
> > For empirical validation, we attach a comparison of the Effective Rank against SimbaV2-SPL.
> >
> > [Graph](https://drive.google.com/file/d/1t3DymR_IOQrFfokIAh7YjfNhcVDD1rOF/view?usp=drive_link)
>
> **Q2. Why is the auxiliary projector considered redundant?**
>
> > Generally, the role of an auxiliary projector in Self-Supervised Learning (SSL) is to preserve the information capacity of the representation. In standard SSL, without a projector, the representation tends to discard all information except what is strictly necessary for the SSL objective. Consequently, information critical for downstream tasks is also discarded, leading to performance degradation [R1].
> >
> > Unlike classification tasks where SSL and downstream objectives differ, the RL+SPL objective intrinsically incorporates $\\mathcal{L}\_{SPL}$​. Thus, requisite information is preserved regardless of R2R2. Furthermore, the added terms ($\\mathcal{L}\_{RR}$​​ and $\\mathcal{L}\_{Var}$​​) explicitly encourage effective dimension utilization without causing information loss. Empirical evidence evaluating an added projector is provided below.
> >
> > [Graph](https://drive.google.com/file/d/1lT5a25FOBOAYDQgiJ4G7LPYGjUdvSF8c/view?usp=drive_link)
> >
> >||+R2R2|+R2R2+Projector|
> >|-|-|-|
> >|Humanoid-Run (UTD=20)| **162.5** | 96.0 |
> >
> > [R1] Florian Bordes, Randall Balestriero, Quentin Garrido, Adrien Bardes, Pascal Vincent: Guillotine Regularization: Why removing layers is needed to improve generalization in Self-Supervised Learning. Trans. Mach. Learn. Res. 2023 (2023)
>
> **Q3. Why does the critic's input include $z'_{t+1}$?**
> >
> > The primary role of the representation obtained from SPL is to distill the core dynamics information of a complex environment and deliver it to the actor and critic. For a similar reason, providing information about the next state as an additional input to the critic is expected to be more advantageous for estimating the dynamics. Empirical support for this has already been demonstrated through an ablation study in the appendix of the TD7 paper [R2] (see Table 9 and Figure 9 in the TD7 paper - Q, remove $z^{(sa)}$). Please note that $z\_{t}^{(sa)}$ in the TD7 paper is equivalent to $z'_{t+1}$.
> >
> >[R2] Fujimoto, Scott, et al. "For sale: State-action representation learning for deep reinforcement learning." NeurIPS 2023.
>
> **Q4. In the lower-left of Fig. 3, should SPL be computed between $z\_{t+1}$ and $z'\_{t+1}$?**
> > Thank you for your careful reading and pointing this out. You are absolutely correct; this was a mistake on our part. We will make sure to correct it in the revised paper.

---

> > ### Author Rebuttal · Reviewer_ausv · 2026-04-04
> >
> > Thank you for the rebuttal. I have no further questions and will carefully consider the final outcome.

---

> > > ### Author Response · Authors · 2026-04-04
> > >
> > > Thank you for the update. I highly appreciate your time and consideration. Please feel free to reach out if any further questions arise in the meantime. Thank you again.

---

### Official Review · Reviewer_up6b · 2026-03-13

**Soundness:** 2
**Presentation:** 3
**Significance:** 4
**Originality:** 2
**Overall Recommendation:** 5
**Confidence:** 4

**Summary:**

This paper studies a specific failure mode of high-UTD off-policy RL with self-predictive learning: representation-level overfitting under intensive experience reuse. The authors propose R2R2, a redundancy-reduction regularizer designed for SPL encoders. The central technical claim is that standard zero-centering, as used in covariance-style redundancy reduction methods, conflicts with the spectral structure of SPL because SPL is argued to preserve a dominant constant eigenmode of the transition operator, while centering removes that component. The main empirical message is that R2R2 improves robustness in high-UTD settings, especially at UTD=20, and that the gains appear complementary to architectural normalization and stronger backbones.

**Compliance With Llm Reviewing Policy:**

Affirmed.

**Final Justification:**

The authors have provided a detailed rebuttal that fully addresses all my concerns.

**Key Questions For Authors:**

Please refer to the weaknesses.

**Limitations:**

Yes

**Strengths And Weaknesses:**

**Strengths.**
This study appears to address a core issue within the literature on Reinforcement Learning with High Update-to-Data (UTD) ratios—one that has rarely been adequately discussed: while most prior methods focus on resolving issues related to value bias or architectural stability, this paper isolates the problem of representational degradation inherent to Self-Predictive Learning (SPL), treating it as a distinct bottleneck. A central concept explored in this work is the inherent tension between redundancy elimination and the "spectral objective" implicitly optimized by SPL. The research not only evaluates the proposed approach across various backbone network architectures but also conducts ablation studies on "zero-centering" techniques and various component loss functions, presenting a comprehensive analysis that combines overall performance curves with singular value diagnostics.

**Weaknesses.**
1. The theory posits that centralization operations eliminate the "constant mode," yet it fails to establish that retaining this mode is *sufficient*—or even *universally beneficial*—for control performance. Since the centering matrix $H = I - \frac{1}{N}\mathbf{1}\mathbf{1}^\top$ has the property of zeroing out (i.e., annihilating) constant vectors, it effectively eliminates the projection of learned representations onto the dominant constant eigenvector of row-stochastic transition operators. The mathematical derivation underlying this specific point is intuitive and straightforward. However, at the interpretive level, the paper makes a far more forceful inferential leap: it moves directly from the premise that "centralization eliminates the constant mode" to the conclusion that "therefore, zero-centering is structurally ill-suited for self-supervised representation learning (SPL) in Reinforcement Learning (RL)." This conclusion lacks adequate evidentiary support.

In Markov analysis or operator theory grounded in spectral analysis, constant eigenfunctions are significant because they reflect the system's stationarity or conservation properties. However, in the context of downstream control tasks, the critical question is whether retaining this mode actually enhances the task-relevant discriminative power of the representation—rather than merely preserving a component that holds a theoretically special status. In fact, the constant mode represents precisely the direction that is *least* informative for distinguishing between different states. Consequently, the paper requires more incisive arguments to explain *why* eliminating this mode in practice degrades the effectiveness of representation learning, rather than simply altering the basis used by the neural network to encode global information. The current argumentation relies primarily on the intuition that the neural network is compelled to implicitly "reconstruct" this information via bias terms; however, this remains merely an intuitive conjecture regarding the optimization process and does not constitute a rigorous proof that such an action necessarily impairs downstream control performance.

2. The motivation for the regularization term is framed as "preserving the dominant mode," yet the proposed non-centering, off-diagonal penalty term does not clearly align with this specific mechanism. This paper replaces centralized covariance regularization with the following formulation:

$L_{\mathrm{RR}}=\frac{1}{d(d-1)}\sum_{i\neq j} [C(Z)]_{ij}^2,\qquad [C(Z)]_{ij}=\frac{1}{N-1}\sum_{b=1}^N z_{b,i}z_{b,j},$

Specifically, a non-centralized correlation penalty term. However, this objective function does not explicitly preserve constant feature modes; it merely avoids subtracting the batch mean while simultaneously suppressing cross-coordinate correlations. These two concepts are not synonymous. Specifically, if a substantial global mean component exists that dominates all coordinates, the non-centralized second-order moments themselves will be heavily influenced by this mean; in such a scenario, penalizing the off-diagonal terms may still force the representation to redistribute or suppress shared global structure—unless that representation happens to align with the coordinate axes in a particularly favorable manner. Consequently, there is a disconnect between the claimed mechanism—"preserving dominant spectral modes"—and the penalty actually imposed—"reducing raw cross-coordinate inner products." The paper's argument would be more compelling if it could derive how this objective function operates within the feature space of the SPL representation, or if it could theoretically demonstrate that the function indeed preserves constant modes while selectively shrinking only redundant correlations. As currently presented, the mechanistic link appears to rest more on assertion than on rigorous argumentation. This potential issue warrants attention, as methods designed to eliminate redundancy—such as Barlow Twins and VICReg—are fundamentally engineered for collapse prevention and decorrelation, rather than for the preservation of specific operator feature modes.

[1] Zbontar, J., Jing, L., Misra, I., LeCun, Y., & Deny, S. (2021). Barlow Twins: Self-supervised learning via redundancy reduction. In Proceedings of the 38th International Conference on Machine Learning (pp. 12310–12320). PMLR.

[2] Bardes, A., Ponce, J., & LeCun, Y. (2022). VICReg: Variance-invariance-covariance regularization for self-supervised learning. In International Conference on Learning Representations.

[3] Tang, Y., Guo, Z. D., Richemond, P. H., Pires, B. Á., Chandak, Y., Munos, R., Rowland, M., Azar, M. G., Le Lan, C., Lyle, C., György, A., Thakoor, S., Dabney, W., Piot, B., Calandriello, D., & Valko, M. (2023). Understanding self-predictive learning for reinforcement learning. In Proceedings of the 40th International Conference on Machine Learning (pp. 33632–33656). PMLR.

3. This paper identifies zero-centering as a critical bottleneck; however, its comparative experiments fail to fully decouple the various contributing factors. Standard methods—such as VICReg and Barlow Twins—typically incorporate several mutually coupled design choices, including mean-centering, normalization schemes, the use of projectors, and the architecture for cross-view alignment. In contrast, the method proposed in this paper eliminates the projector alongside the mean-centering step, subsequently comparing this configuration against a zero-centered variant in ablation studies. While this comparison offers some informative value, it remains unable to conclusively isolate the true source of the observed performance gains: do they stem from (i) the avoidance of mean-centering? (ii) a better alignment with the optimization geometry of SPL achieved by directly regularizing the encoder? or simply (iii) a reduction in the optimization difficulty associated with high-UTD training?

---

> ### Author Rebuttal · Authors · 2026-03-31
>
> Thank you for your in-depth review and sharp insights. We apologize for any lack of clarity in our paper.
>
> **W1. Is the constant mode really that important?**
>
> > SPL representations compress complex dynamics to directly inject hard-to-infer structural information into RL networks, extending beyond mere state distinction.
> >
> > The true value function satisfies the Bellman Equation: $V^\\pi = R^\\pi + \\gamma P^\\pi V^\\pi$. Solving for $V^\\pi$ via the Successor Matrix $M = (I - \\gamma P^\\pi)^{-1}$ yields:
> > $$V^\\pi = M R^\\pi$$
> >
> > Eigendecomposing the transition matrix $P^\\pi = \\sum_i \\lambda_i u_i v_i^\\top$, $M$ shares identical eigenvectors:
> > $$M = \\sum_i \\frac{1}{1 - \\gamma \\lambda_i} u_i v_i^\\top$$
> >
> > Since $P^\\pi$ is a row-stochastic matrix, the Perron-Frobenius theorem dictates its largest eigenvalue is strictly $\\lambda_1 = 1$, with the corresponding right eigenvector $u_1 = \\mathbf{1}$ (the Constant mode). For the remaining eigenvalues, $|\\lambda_i| < 1$. Substituting this yields:
> > $$V^\\pi = \\left( \\frac{1}{1-\\gamma} \\mathbf{1} v_1^\\top + \\sum_{i \\ge 2} \\frac{1}{1 - \\gamma \\lambda_i} u_i v_i^\\top \\right) R^\\pi$$
> >
> > With $\\gamma$ typically set near 0.99, the Constant mode's weight is $\\frac{1}{1 - 0.99} = 100$. This massively dominates the remaining modes ($\\frac{1}{1 - \\gamma \\lambda_i} < 100$). Thus, within the target $V^\\pi$ space, the Constant mode ($\\mathbf{1}$) is the overwhelmingly dominant principal component.
> >
> > Erasing the constant mode ($u_1 = \\mathbf{1}$) via zero-centering fundamentally distorts the encoded dynamics' spectral structure. While a Critic could theoretically recover this global shift using a bias term, removing this structural anchor leaves only transient dynamics ($u_2, u_3, \\dots$), forcing an over-reliance on volatile, high-frequency features during bootstrapping.
> >
> > Although feeding both $z$ and $s$ to the critic (e.g., in TD7 and SimbaV2-SPL) mitigates this issue, $z$ is designed to accelerate convergence by providing a structurally sound representation. Discarding its dominant spectral component inherently compromises this goal.
> >
> > Empirically, the graph below tracks representation Effective Rank (ER) during training, comparing zero-centered and uncentered (R2R2) variants on Humanoid-Run (UTD=20):
> >
> > [Graph_ER](https://drive.google.com/file/d/1WnrX4kRdWbhVad_UjjFaufOl8Qu8KJNK/view?usp=drive_link)
> >
> > [Graph_Return](https://drive.google.com/file/d/11HUg6zBtJSeqjwy-H15jCfkGzSixtSCP/view?usp=drive_link)
> >
> > As observed, the zero-centered variant exhibits a noticeably lower ER. This result indirectly supports our hypothesis: explicitly removing the constant mode may force the network to unnaturally compensate using the remaining features. This likely induces feature co-adaptation and partial dimensional collapse, manifesting as the tail cut-off in Figure 6 (Right) of our main paper.
>
> **W2. How the Objective Function Operates Within the Feature Space?**
>
> > We agree that preventing mean subtraction and preserving the constant mode are not strictly synonymous: minimizing non-centered cross-terms in $\\mathcal{L}_{RR}$ could hypothetically drive the network toward (1) suppression ($\\mu \\to 0$) or (2) redistribution (orthogonalizing remaining representations).
> >
> > However, our framework structurally avoids suppression and enforces redistribution through two key mechanisms:
> >
> > **i. Joint Optimization with $\\mathcal{L}\_{SPL}$:** The $\\mathcal{L}\_{SPL}$ objective inherently attempts to capture the dominant constant eigenvector, maintaining a non-zero bias ($||\\mu|| > 0$). Jointly optimizing $\mathcal{L}\_{SPL}$ and $\mathcal{L}\_{RR}$ severely penalizes trivial suppression, forcing the network to redistribute the constant energy while orthogonalizing the remaining features.
> >
> > **ii. Architectural Constraints:** L1/L2 normalization layers in TD7 and SimbaV2-SPL prevent representation magnitudes from shrinking to zero, explicitly eliminating the suppression shortcut.
> >
> > Figure 6 (right) demonstrates this: TD7+R2R2 maintains a high Effective Rank (ER) without tail cutoff, indicating successful orthogonalization rather than trivial suppression.
>
> **W3. Addressing the Source of Performance Gain**
>
> > R2R2's performance gains primarily stem from direct regularization, which shapes spectral properties toward an ideal distribution (as discussed in Section 5.6). Additionally, avoiding zero-centering yields a separate performance boost (Fig. 5, left).
> >
> > To address the insufficient ablation on further design choices (e.g., projector usage), we provide supplementary results below:
> >
> > [Graph](https://drive.google.com/file/d/1fHvJU1Vhv8rP6CcV18dywylPHSxKf8by/view?usp=drive_link)
> >
> >| +R2R2 | +R2R2 (VICReg->BT) | +R2R2+Projector | +R2R2+Zero-Centering |
> >|-|-|-|-|
> >|**162.5**| 40.6 | 96.0 | 78.3 |
> >
> > Evaluated on Humanoid-Run at UTD=20, a highly challenging regime, the empirical results confirm that R2R2 achieves the best performance.

---

> > ### Author Rebuttal · Reviewer_up6b · 2026-04-02
> >
> > The author's response has fully addressed all my questions. I will raise my score accordingly. Thank you for helping me gain a thorough understanding of this work.

---

> > > ### Author Response · Authors · 2026-04-03
> > >
> > > We sincerely appreciate your insightful review and the upgraded recommendation score. Your feedback allowed us to incorporate valuable additions that significantly enhance the persuasiveness of our manuscript. Thank you once again for your time and valuable review.

---

### Decision · Program_Chairs · 2026-04-30

**Decision:**

Accept (regular)

**Comment:**

A technically strong paper with consensus among the reviewers.